# Consecrating the Peripheral: On the Ritual, Iconographic, and Spatial Construction of Sui-Tang Buddhist Corridors

Zhu Xu 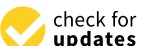

School of Architecture, Harbin Institute of Technology, Shenzhen 518055, China; xuzhu@hit.edu.cn

**Abstract:** The corridor-enclosed cloister characterized Buddhist monasteries during the Sui and Tang periods. This architectural form was first introduced by Emperor Liang Wudi from the palace and continued to prevail until the eleventh century, when a gradual transformation occurred, resulting in the corridor evolving into a long, narrow image hall. This paper examines the ritual and pictorial programs of the Sui-Tang Buddhist corridor to gain insight into this transformation and its ceremonial significance. Specifically, it explores how the corridor was empowered by the state-sponsored maigre feast as a place of worship and how the monastic community of a particular school appropriated the space to celebrate an unbroken dharma-transmission lineage from the Buddha to a specific group of Chinese patriarchs. Lastly, the paper aims to comprehend the adaptation of the corridor into an image hall, which was influenced by political and religious shifts in the eleventh century when Buddhist monasteries were no longer designated as the ritual arena for the state-sponsored maigre feast.

**Keywords:** Buddhist corridor; paintings of divine monks; maigre feast; Sui and Tang

## 1. The Sui-Tang Buddhist Corridor: Understanding the Ritual-Architectural Transformation of the Peripheral Structure in a Medieval Chinese Monastery

In examining the architectural transformation of Buddhist monasteries between the Sui-Tang and Liao-Song periods, scholars often draw attention to the enclosing structures surrounding the courtyard compounds (Z. Xu 2020, p. 53). This is because these structures underwent a significant transition from four-sided open corridors—colonnaded arcades wrapping around a courtyard—in the 7th century to two-sided semi-open verandahs in the 11th century, which reflects a fundamental ritual shift, as the Sui-Tang courtyard-centric public ceremonies, where corridors played a crucial role, were replaced by Liao-Song indoor worship conducted within long, narrow halls beneath the eaves of verandahs (Z. Xu 2016, pp. 105–16). The corridor was initially introduced into Chinese Buddhist monasteries in the early 6th century as part of the palace architectural model, specifically designed for the emperor's grand dharma assemblies. In this context, its primary function was to accommodate the congregation of scholar monks rather than serve as a space for worship. A key research question thus arises: when and how did the ritual-architectural transformation that led to the *closing* of the corridor begin?

To address this question, we must investigate the iconography, if any, of Sui-Tang Buddhist corridors, as this could indicate the corridors' devotional function. Despite the lack of physical or visual evidence, a thorough examination of Tang and early Song textual materials provides information from twenty-three urban monasteries (Table A1). It is notable that the back walls of the Buddhist corridors served as a canvas for mural paintings depicting various Buddhist themes, the majority of which featured portraits of monks. This article aims to explore the often-overlooked spatial and ceremonial significance of Sui-Tang Buddhist corridor paintings. The discussion is organized into three sections. The first section uncovers the worship of divine monastic beings as the primary catalyst for the emergence of Buddhist corridor paintings and provides a thorough historical analysis

of how the connection between the corridor and divine monastic beings was established within the specific ritual context of Tang imperial practices. The second section seeks to reconstruct the pictorial programs of corridor paintings in several Tang monasteries using textual records from Zhang Yanyuan's 張彥遠 *Lidai minghua ji* 歷代名畫記 (Records of Famous Paintings of All Dynasties), Duan Chengshi's 段成式 *Youyang zazu* 酉陽雜俎 (The Miscellany from Youyang), and the catalogues and diaries of ninth-century Japanese pilgrims. The final section delves into the monastic ritual shift and the evolving role of Buddhist monasteries within the Song state sacrificial system, shedding light on the decline of corridor paintings and the associated transformation of corridor-verandah structures.

## 2. Portrait Gallery: The Introduction of Wall Paintings

No evidence exists to indicate the presence of Buddhist corridor murals in the Northern and Southern dynasties. Although Emperor Liang Wudi 梁武帝 (r. 502–549), who pioneered the use of corridors in monasteries, had commissioned wall paintings by Zhang Sengyou 張僧繇 to adorn monasteries, all of the artist's known works were created inside halls rather than corridors (Zhang 2018, p. 160). This argument gains further support from the observation that Japanese monasteries constructed during the Asuka period, which were modeled after the Liang Buddhist culture, also lack corridor paintings (Uehara 2021). In textual sources, Buddhist corridor paintings from the early years of the Sui dynasty serve as the earliest examples (no. 1–5, Table A1), indicating that such a tradition emerged around a century after its initial introduction. On the other hand, early Tang writings on monastery design, such as Daoxuan's *Zhong Tianzhu Sheweiguo Qihuansi Tujing* 中天竺舍衛國祇洹寺圖經 (Illustrated Scripture of Jetavana Vihāra of Śrāvastī in Central India), imply that corridor paintings may not have been widely adopted by the mid-seventh century. This scripture, which reflects the ideal conception and actual construction of the early-Tang state monastery, does not depict the corridors of the Central Buddha Cloister 中佛院 as a place of veneration. The only example of a corridor mural found in the text is at the Cloister of Impermanence 無常院, where paintings of white bones are prepared as a special deathbed ritual for dying monks (Daoxuan 1924–1933g, 893c10–12).[1] The two sources are not contradictory, as it is reasonable for a newly emerging tradition to take several decades to fully develop and gain widespread acceptance. Nevertheless, the growing number of reports on Buddhist corridor paintings between the seventh and ninth centuries, not only in Chang'an but also in other major cities of China, provide compelling evidence of the expanding popularity of this religious art form. Reaching its peak in the early eighth century, this practice was brought to Japan by pilgrim monks and praised as an innovative, unparalleled style when employed in the reconstruction of the state monastery, Daianji 大安寺, in Nara (Uehara 2021; Wong 2018, pp. 154–61).

Remarkably, all the known Sui Buddhist corridor paintings share the subject of depicting portraits of monks. This theme continued to be prominent throughout the Tang dynasty, featuring a group of legendary, imaginary, and divine monastic figures arrayed on the long walls of corridors and variously identified as *shengseng* 聖僧 (holy monk), *gaoseng* 高僧 (eminent monk), *xiansheng* 賢聖 (sage and saint), *luohan* 羅漢 (arhat), or *zushi* 祖師 (patriarch). Understanding the Buddhist corridor as a monastic memorial space requires further examination, as the driving forces behind this practice remain unclear. This is especially intriguing, considering that portrait halls (*yingtang* 影堂) were often built in Tang monasteries for the formal commemoration and veneration of eminent monks. Why is a corridor the proper place to accommodate this ritual activity?

### 2.1. Windowed Corridor: The Tradition of Non-Buddhist Ceremonial Compound

However, the ceremonial adaptation of the enclosing corridor as a portrait gallery was unlikely to be derived from a pre-Tang tradition of palace, ritual, and administrative architecture. The practice of creating portraits to honor historical and contemporary personages began in the Han dynasty (Seckel 1993). Many documented cases, according to a wealth of textual information and even some original works that have been luck-

ily preserved, were painted on the walls of palace halls, governmental offices, as well as funeral offering shrines.[2] However, none of them were found painted in corridors or corridor-like structures.

On the other side, despite limited information from archaeological and visual materials, insights into the design of early enclosing corridor structures can be drawn from the Zhaoyang Hall 昭陽殿 compound, the main audience building of the Northern Qi palace built in 539 at Ye. A mid-Tang source describes its eastern and western corridors as having elongated windows, likely installed along the rear walls, to accommodate court musicians during ceremonies.[3] The presence of windows strongly indicates that the corridors may not have significant pictorial programs. Between the late-sixth and early-seventh centuries, the extensive use of windowed corridors in religious and palatial compounds is evidenced by the creation of the term *xuanlang* 軒廊 ("windowed corridor"), coined in early Tang literature to describe this distinctive architectural element.[4] The seventh-century Daoist monastic code, *Sandong fengdao kejie* 三洞奉道科戒 (Rules and Precepts for Worshipping the Dao), includes guidelines for monastery buildings in Section 4 *Zhiguanpin* 置觀品 (Setting Up Monasteries). It stipulates that a *xuanlang* structure should be built for passageways and circumambulation around sanctuaries, halls, pavilions, and terraces within a cloister, encircling all four sides (Kohn 2004, p. 95). Built in 605, Qianyang Hall 乾陽殿, the main audience building of Emperor Sui Yangdi's 隋煬帝 Ziwei Palace 紫微宮 at Luoyang, along with several other significant ceremonial-sacrificial compounds within the palace, were characterized by a *xuanlang*-enclosed design (Wei and Du 2006, pp. 8–10). Arguably the best example of the *xuanlang* structure, although located outside China, is the early eighth-century corridor that encloses the western compound of Hōryūji 法隆寺 in Ikaruga prefecture, Japan (Figure 1).

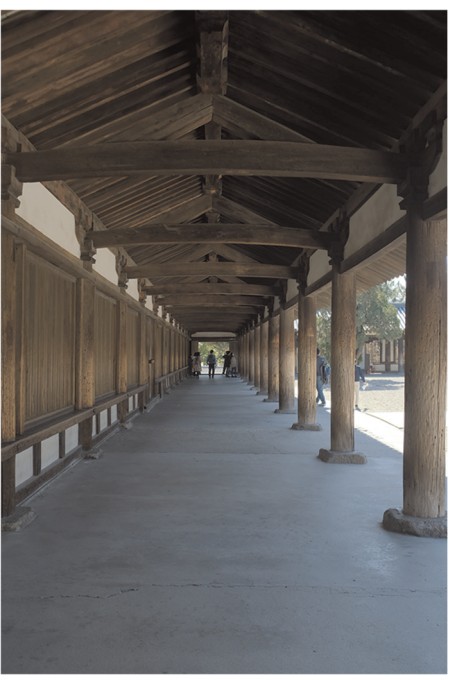

**Figure 1.** Windowed Corridor (*xuanlang*) at Hōryūji, Ikaruga, 670–747 (the author's photo).

*2.2. The State-Sponsored Maigre Feast: Incense-Procession in the Monastic Corridor and the Cultic Worship of Divine Monastic Beings*

Whereas the Sui-Tang Buddhists corridor paintings probably did not stem from a pre-existing tradition, a close examination of the corridor's ceremonial significance would reveal an association with the cult of divine monastic beings. This association could potentially lead to the corridor's consecration as a space appropriate for venerating holy



monks through portraits, which is the earliest and most popular kind of painting scheme in the corridor.

The cult of divine monastic beings can be traced at least as far back as Dao'an 道安 (314–85) and Huiyuan's 慧遠 (334–417) worship of the arhat Piṇḍola (Ch. 賓頭盧), by inviting him to maigre feast (*zhaihui* 齋會)–the lay-monastic assemblies for offering vegetarian meals to monks–and presenting him with food and/or a bath.[5] From that time until the Tang dynasty, it became customary during the maigre feast to prepare an empty seat in the midst of the monastic congregation for Piṇḍola or any other divine beings about to receive food offerings. The arrangement led to a gradual yet substantial development of spatial consecration, as the location where monks sat to receive food offerings also came to be considered as a site for encountering divine monastic beings. Indeed, for medieval Chinese Buddhists, any monk invited to the maigre feast could be a hidden divine being, as the arrival of the divine was mysterious, with their presence either invisible or disguised as that of an ordinary individual. This is best illustrated by the miraculous arrival and vanishing of a mysterious monk at the imperial-patronized maigre feast for two hundred monks in 460, held at Zhongxing Monastery 中興寺 in Jiankang (Huijiao 1924–1933, 372c10–15).

In addition to setting up empty seats, Dao'an and Huiyuan also introduced the performance of *xingxiang* 行香 into the ceremonial program of the maigre feast for the worship of divine monastic beings (P. Wang 2020). *Xingxiang*, also known as incense-procession prayer, is an offering practice popular in medieval China. At the beginning of a ceremony, the patron or superintendent would circumambulate the ritual arena holding an incense burner, which is intended to invite holy and ordinary monks from all ten directions to receive alms.[6] The practice of *xingxiang* further reinforced the connection between the space where monks gathered for feasting and the space dedicated to worshiping divine monastic beings. As a result, the increasing cultic worship of divine monastic beings led to the production of the earliest known portrait images of these beings in 470–471 by monks from the capital monasteries at Jiankang (Daoshi 1924–1933, 609c9–10).

The week-long prayer ritual of a maigre feast for the well-being of Emperor Qi Wudi 齊武帝 at 490 is arguably the most renowned divine monk worship ceremony ever recorded in the history of Southern and Northern dynasties (Daoshi 1924–1933, 609c13–20). Performed in Yanchang Hall 延昌殿, the Emperor's formal residence within the imperial palace, which temporarily served as the feast chamber (*zhaishi* 齋室), this event featured offerings of vegetarian meals and incense, and the practice of *xingxiang* was undoubtedly employed, as evidenced by the specific mention of the censer. The Yanchang Hall feast serves as a unique and extreme example of a maigre feast, where the ceremonial space was fully dedicated to the cultic worship of the Buddha and divine monastic beings. Interestingly, its ritual program did not significantly deviate from those attended by ordinary monks. Although not explicitly recorded, the presence of portraits of divine beings likely served as visual objects of veneration within the hall, given that such practice, as previously mentioned, had emerged at Jiankang for divine monk worship decades earlier.

Emperor Liang Wudi's innovative monastery design incorporated the corridor as a "space for monks", as evident in the ceremonial plan of the grand dharma assembly at Tongtai Monastery 同泰寺, where the corridor was designated for seating senior scholar monks. Given the significant role of the maigre feast in the grand dharma assembly and the emperor's reputation as a fervent Buddhist promoting the cultic worship of divine monastic beings[7], one might reasonably expect an elaborate pictorial program related to divine monks within the corridor space. However, textual records are silent on this matter. A possible explanation could be that the emperor's religious and political agenda was to project imperial spatial order into the Buddhist realm. The corridor in the imperial palace compound, which served as the model for the Buddhist corridor, functioned as a space for low-ranking officials to play court music during the state rite of grand audience. Thus, the Buddhist corridor primarily functioned as a space where scholar monks received teachings from the emperor, serving as agents of imperial authority. Any consideration given to the veneration of divine monastic beings, if present at all, could not overshadow this overarching symbolism. Moreover, in the

grand dharma assembly, vegetarian meals were widely provided to all people in the capital to showcase the emperor's persona as a generous bodhisattva. In this context, the corridor–even though monks likely took their meals there–maintained its basic status as an ordinary dining zone, being part of the emperor's "universal offerings (*biangong* 遍供)", rather than a space specifically designated for anticipating the arrival of divine beings. In summary, the emphasis on the cult of divine monastic beings in the emperor's grand dharma assembly was significantly less than in earlier maigre feast practices, mainly due to its unique ritual agenda, which was closely tied to the imperial presence.

Nonetheless, the Liang imperial ceremonial plan of placing monks in the corridor space persisted, along with the widespread adaptation of the Liang monastery architectural-ceremonial model throughout China. Despite the scarcity of records from the sixth and seventh centuries, there are some details available on the Sui arrangement of the maigre feast. One example is the special event held in the imperial palace to celebrate the emperor's construction of thirty śarīra stūpas across China in 601 (Daoxuan 1924–1933b, 214b9–13). The maigre feast, which served as the closing ceremony of this event, took place in the eastern corridor of the main audience compound. Here, Emperor Sui Wendi 隋文帝 (r. 581–604), accompanied by all civil and military officials, partook in vegetarian meals. This event can be reasonably considered to have no fundamental difference from those staged in the Sui monastery, as this period saw the central cloister of a state monastery share many architectural characteristics with the main audience compound of the imperial palace.

During the same period, a notable change occurred as the maigre feast, which was essentially an act of offering to gain merit, once again became the central ritual in the imperial Buddhist assembly. This transformation was primarily due to the growing devotional role of imperial Buddhist practices. Unlike Emperor Liang Wudi, who had given public dharma lectures personally, very few secular rulers could do the same. Consequently, their primary role transitioned to leading the offering worship for the maigre feast. Emperor Sui Wendi was one of the most renowned practitioners, with monastic texts documenting his frequent performance of the *xingxiang* ritual, where he would hold the censer and lead the procession during the regularly held maigre feast at Daxingshan Monastery 大興善寺, the supreme state monastery of the Sui dynasty (Daoxuan 1924–1933f, 437a20–23). Naturally, the worship of divine monastic beings, which forms the devotional foundation of the medieval maigre feast, would be emphasized in Sui practice and spatially linked to the feast's location in state ceremonies, such as the corridor. This ritual-spatial connection is intriguingly reflected in a late-sixth-century Northern Qi miracle tale, which narrates a monk's visit to a divine monastery and his encounter with the arhat Piṇḍola in the western corridor of the monastery's central cloister.[8]

The last significant development, and likely the most critical factor contributing to the consecration of the corridor, was the integration of the maigre feast into the Tang state sacrificial system, for the purpose of praying for the afterlife well-being of the deceased emperors. Adding Buddhist rituals in service of national mourning dates back to 582, when Emperor Sui Wendi ordered the construction of Buddhist monasteries in four cities. On the anniversary of his father's death, these monasteries were to conduct a series of rituals, including the maigre feast, image making, perambulation (Skt. *Caṅkrama*) or circumambulation (Skt. *Pradakṣiṇā*), and repentance through observing the eight precepts (Fei 1924–1933, 107b29–c15). The Tang rulers gave this imperial practice greater emphasis. By 727, this ceremony had been enacted nationwide and was subsequently codified into the administrative law, i.e., *Tang Liu Dian* 唐六典, as a government-hosted observance (Li 1992, p. 217). Known as *guoji xingxiang* 國忌行香 (Incense-Procession Prayer Ceremony on the National Memorial Day), it was prescribed to be performed annually in official monasteries across eighty-one prefectures throughout China.[9]

For details of the Tang *guoji xingxiang* ceremony, the most meticulous account is left by the Japanese pilgrim monk Ennin 円仁, who visited China in the mid-9th century and personally encountered the ceremony taking place at the Kaiyuan Monastery 開元寺 in Yangzhou 揚州, on the anniversary of Emperor Jingzong's 唐敬宗 death in 838. Ennin reports this cer-

emony as a maigre feast for five hundred monks, with the core offering ritual—*xingxiang*—performed by the Minister of State (hereafter, Minister) and Commander-in-Chief (hereafter, Commander) on behalf of the emperor. According to the description, a theoretical reconstruction of the monastery's ritual-architectural plan has been developed, featuring a corridor-enclosed compound with a main gate, a central gate, a Buddha hall, and a lecture hall aligned along the central axis (Figure 2). In the early morning, the five hundred monks seated themselves in rows on the northern, eastern, and western sides of the corridor[10], awaiting the commencement of the ceremony. Once the Minister and Commander were prepared.

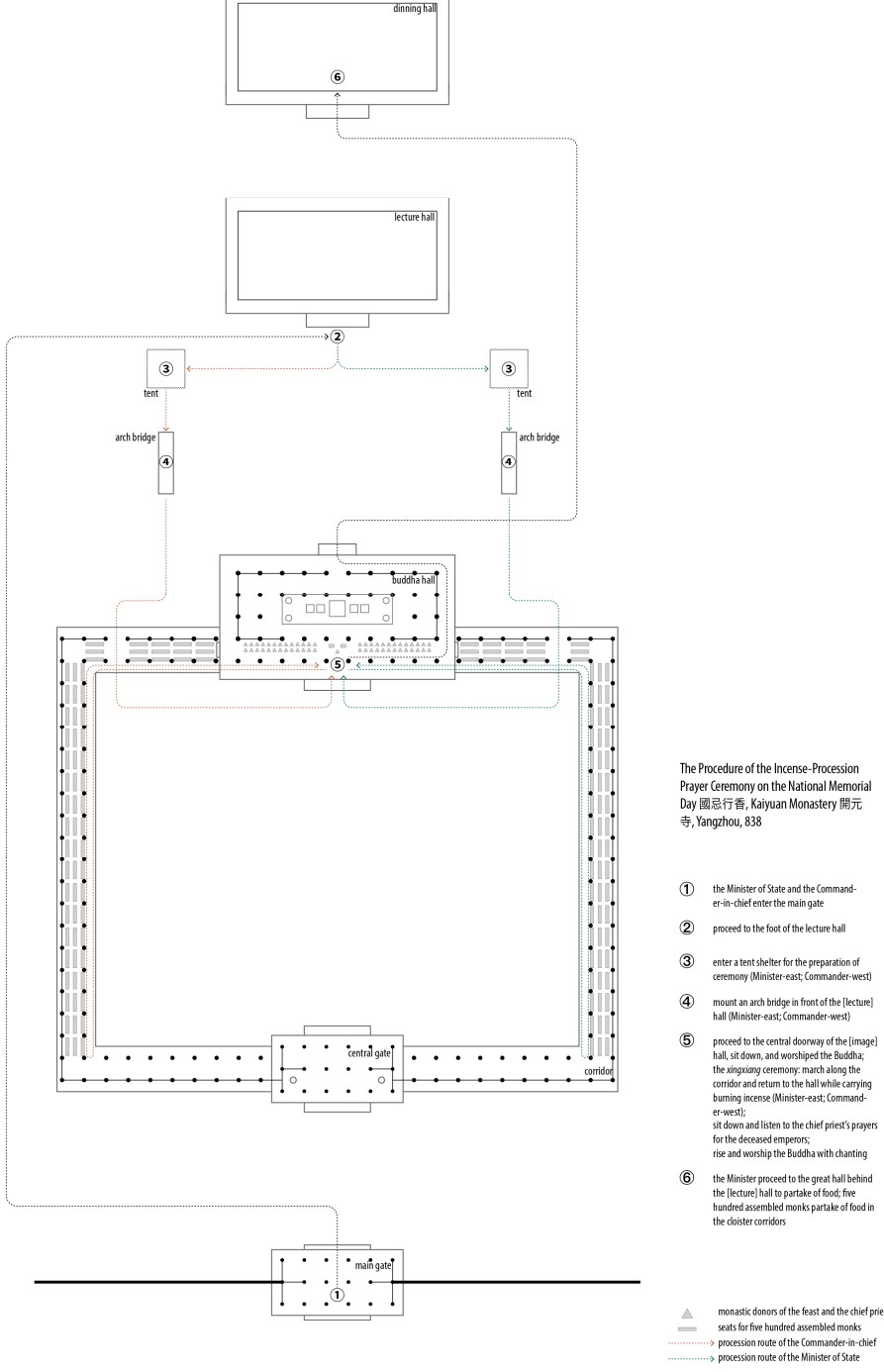

**Figure 2.** Kaiyuan Monastery, Yangzhou, the *guoji xingxiang* ritual performed the eighth day of the twelfth month, 838. (author's reconstruction).

They met at the center doorway of the (Buddha) hall.[11] There they sat down and worshipped the Buddha; this done, several tens of monks stood in rows at the eastern and western doorways of the hall (respectively), each holding a lotus flower and a green banner. A monk struck a gong, and chanted the general formula of humility, adoration, and steadfast devotion to the Three Treasures. This being completed, the Minister and the Commander rose to their feet and took censers. The district officials ranged themselves behind them, distributing censers. Then the two lines marched to the east and west. The Minister proceeded eastward; the monks who were carrying flowers and banners led the way, chanting in unison the two-line Sanskrit *gatha*, 'the marvelous body of the Tathāgata,' etc. First, (after these) came an old priest of great holiness, and then the soldiers followed as bodyguards. They went to the foot of the cloister corridor eaves, and all the monks (seated in the corridor) received the incense-procession offering. This being completed, they turned around and took the same route to the hall, chanting uninterruptedly in Sanskrit. The Commander went through the incense-procession ceremony on the west with the same forms as those observed on the east. The two came back to their starting points at the same time, and at that moment, the mingling of voices in the chanting (by the group of monks) on the eastern and western doors of the hall was most wonderful. During all this, the hymn leader had not moved but stood alone, striking the gong. At a pause in the Sanskrit chanting, he again intoned the formula of adoration and steadfast devotion to the Three Treasures. The Minister and the Commander both sat down at their original places, each with two incense burners that he had received at the time of the incense ceremony. An old priest of great holiness, Yuancheng, read the prayer for the occasion. That being completed, the hymn leader intoned the stanzas praising the Eight Classes of Demi-gods. The purpose of the wording was to glorify the spirit of the late emperor. At the end of each phrase, repeat the formula of adoration and steadfast devotion to the Three Treasures. The Minister and the various civil officers rose together and worshipped the Buddha three or four times as they wished with chanting. The Minister and the rest, led by soldiers, proceeded to the great hall behind the (Buddha) hall and went inside to partake of food. The congregation of the five hundred assembled monks partook of food in the cloister corridors.[12]

Ennin's careful record provides us with a wealth of information regarding how the Buddhist corridor was envisioned to satisfy the Tang performance of the *guoji xingxiang* ceremony. This also allows us to speculate on the Sui organization of the imperial assembly at Daxing-shan Monastery. Although the corridor space primarily functioned to house the five hundred assembled monks, the processional offerings—including incense, flowers, banners, and chanting—underscore the imperial veneration of the Buddhist church and ritually mark the corridor as a space of worship. Moreover, when the Minister and the Commander proceeded along the corridor, their offerings were dedicated to not only the assembled monks but also to divine monastic beings. Ennin mentioned in his diary, from the entry for the Lantern Festival Day at 839, that monks of the Kaiyuan Monastery offered oil lamps in the corridor as a way to worship the portraits of patriarchs (*shiying* 師影) (Ennin 2007, p. 97). Viewed from the standing point of the procession, which was at the outermost position of the corridor, the assembled monks and the portrait paintings behind them were visually connected, blending the earthly and the divine into one community of sangha. This arrangement is likely brought about by the ritual-architectural interplay between the maigre feast and corridor under the imperial imagery of Buddhist religious space. On the one hand, the worship of divine monastic beings, integral to the maigre feast, was greatly emphasized due to the imperial practice that anticipated an increased incorporation of cultic worship. On the other hand, this cultic worship was envisioned to be staged within the corridor-enclosed central cloister of a monastery, ensuring that the grandeur and solemnity of the imperial authority were appropriately projected. Through comparison with the conventional performance of the maigre feast by non-imperial members of the laity and sangha, it becomes more evident that the *guoji xingxiang* was indeed a unique, highly imperial-style event. Our understanding of the conventional Tang maigre feast is enriched by Ennin's account, as he himself sponsored a maigre feast in the same Kaiyuan Monastery on the anniversary of the death of Tiantai Master Zhiyi 智顗 (Ennin 2007, pp. 69–71). This ceremony, which followed almost every step of ritual activity

in line with the *guoji xingxiang*, was notably held in the dining hall rather than the corridor-enclosed central cloister.

Interestingly, the impact of state-sponsored maigre feasts on the rise of corridor paintings was not solely due to the inherent worship of divine monks but also in relation to the performance itself. The correlation is due to the fact that the practice of arranging monks seated in the corridor for lunch, under the ceremonial and political significance of these feasts, soon became iconic imagery representing the ceremony. This can be observed in several eighth- and ninth-century illustrations of the Medicine Buddha Sūtra 藥師經 and the Vimalakīrti Sūtra 維摩詰經 at Mogao Grottoes 莫高窟 (Figure 3).[13] In fact, this particular practice had a far-reaching influence, as its application even extended beyond the Buddhist religion, creating a model for the Tang court ritual to follow. In 630, an imperial edict was issued, requiring the provision of lunch meals in the outer corridor for officials after their attendance at the daily court audience. Known later as "dining-in-corridor" (*langxiashi* 廊下食), this practice was established as a court ritual order, designated to be performed under the porch of the Deliberation Halls (*chaotang* 朝堂).[14] These halls were built to serve as places for officials to discuss government affairs and await the commencement of court audiences. As long structures situated symmetrically to the west and east in front of the main gate hall in Taiji Palace 太極宮 and the main audience hall in Daming Palace 大明宮, these buildings were reminiscent of corridors in the central cloister of a monastery. While it is clearly untrue that the painted scenes of seated monks in the corridor were directly responsible for developing the corridor paintings in physical space, the fact that these scenes were singled out to represent maigre feasts hints that this particular practice was perceived as a pictorial theme in connection with the Buddhist corridor.

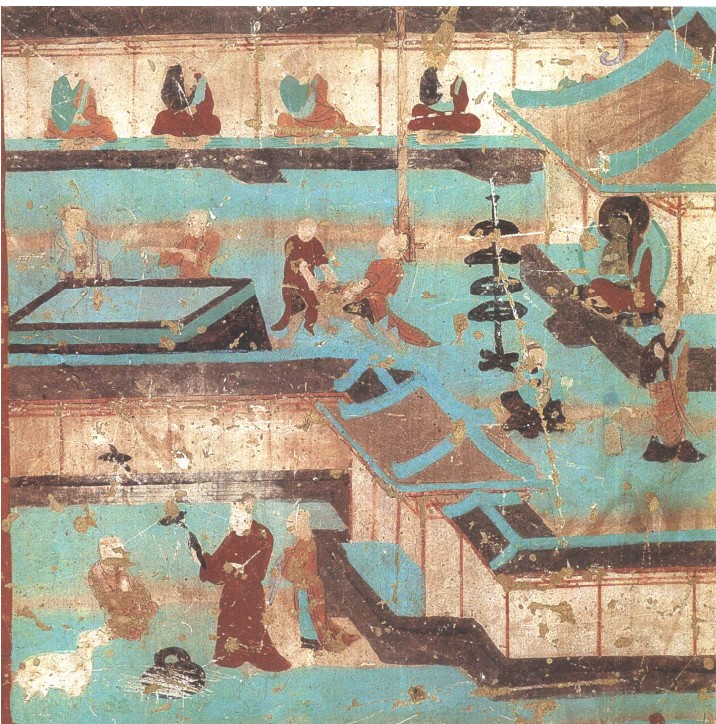

**Figure 3.** A Scene of Maigre Feast in a Buddhist Monastery, Illustration of Medicine Buddha Sūtra, ninth century, Northern Wall of Cave 12, Mogao Grottoes (Tan 1999, Figure 183).

### 2.3. Offerings of Performing Entertainment in the Corridor-Enclosed Courtyard

One more significant yet undiscussed ritual advancement that anticipated the corridor's consecration is the use of the monastic courtyard to stage the performance of court banquet music and diverse forms of entertainment. This celebratory ritual was by no means part of the solemn mourning of the *guoji xingxiang*, but it was a notable component of grand Tang Buddhist festivities.

Ceremonial performances of court music within a Buddhist monastery can be found in records of early Tang activities, with the earliest example being the inauguration ceremony of the imperial-sponsored Great Ci'en Monastery 大慈恩寺 in 648. In the Biography of the Tripitaka Master Xuanzang 玄奘, it is described that this grand event started with a morning procession, characterized by bejeweled chariots carrying fifty distinguished monks and numerous Buddhist images, scriptures, and sacred objects along the capital's main thoroughfare. The procession was accompanied by thousands of laity and monastics from the capital, along with civil and military officials holding incense and flowers, and chanting praises. Court bands of Nine-Part Music (*jiubu yue* 九部樂) were also present with the performance of various other forms of entertainment. When the procession reached the monastery gate, the Duke of Zhao and the Duke of Ying, together with the Chief of the Imperial Secretariat, were ordered by the emperor to receive and place the images and scriptures in the main hall, while holding censers in their hands. The bands played the Nine-Part Music, and dancers performed the Dance of Triumph (*pozheng wu* 破陣舞) and other acrobatic feats in the courtyard (Huili 1995, pp. 218–20). The Nine-Part Music is the most widely performed banquet entertaining music (*yanyue* 燕樂) of the early Tang imperial court, which consisted of nine troupes, each responsible for the performance of a particular group of music and dances. The display and performance of the Nine-Part Music in the monastic courtyard became a model for imperial Buddhist ceremonies, as evidenced by several later examples found in the Great Ci'en Monastery and other great capital monasteries during the seventh and early eighth centuries.[15]

In the Tang celebrations, monastic courtyards served as vibrant venues for music and dance performances, while vegetarian feasts, a prevalent offering act in medieval China, could also occur within the same cloister. This practice is seen as early as 656, when the Nine-Part Music performance and a maigre feast were organized to entertain two thousand monks at the Great Ci'en Monastery in commemoration of an imperial gift. Over time, it evolved into a tradition and was eventually incorporated into the state ritual code. In 838, the Central Secretariat (*Zhongshu sheng* 中書省) and the Chancellery (*Menxia sheng* 門下省) reported that, as part of the annual celebration of the emperor's birthday, officials stationed in the capital were required to visit great monasteries and sponsor thousand-monk feasts therein. Additionally, court music performances were to be held simultaneously to honor their offering of the incense procession ceremony (Wang 1773, p. 1470).

Entertaining performances in service of Buddhist ceremony, as evidenced by the depiction of dancing music offerings (*jiyüe gongyang* 伎樂供養) in the second-century translation of *Daoxing bore jing* (Aṣṭasāhasrikā Prajñāpāramitā Sūtra 道行般若經), had a long-standing tradition in India and Central Asia, and was known by the Chinese through the introduction of Buddhism. Early practices in China were seen in the mid-fifth century, first in the celebrations of the Buddha's birthday.[16] During this festival, images and statues of the Buddha are paraded in chariots through the streets, accompanied by musical, dancing, and acrobatic performances.[17] Several decades later, music and dances began to be staged within monasteries in early-sixth-century capital cities. For example, the Jingle Nunnery 景樂尼寺 and Wangdianyu Monastery 王典御寺 in Northern Wei Luoyang provided such performances on the six monthly fast days (*liuzhairi* 六齋日) in the courtyard in front of the Buddha hall (Yang 2000, pp. 42, 134). In the south, Emperor Liang Wudi composed a set of ten hymns called "orthodox music (*zhengyue* 正樂)", and commissioned them to be performed in the form of dances, chants, and songs during his great dharma assemblies to convey Buddhist teachings (Wei 1973, p. 305). While little is known about the spatial organization of pre-Tang musical-dancing performances in monastic courtyards, they were unlikely grand and sophisticated settings akin to those found in Tang Buddhist ceremonies. Such transformation could only occur when state ceremonial music was restructured to be appropriate for use in a Buddhist environment. During the Northern and Southern dynasties, the music and dances of Chinese court ceremonies, including those for banquet entertainment, were associated with statecraft and Confucianism. Nevertheless, the increasing popularity of foreign-style music in the same historical period provided material background for the development of a richer form of court music, and led to a groundbreaking shift in the use of non-Chinese musical instruments and

repertory for court banquet entertainment.[18] Collectively, a group of music from bordering countries, consisting of five, seven, or eight parts, was further organized with Chinese music into a combinatory system from the early Sui period, and was displayed and performed in a predetermined order during the most important court banquet occasions. One notable characteristic of these foreign musics, particularly those from India and Central Asia, was their close Buddhist affiliations, as the diffusion of Buddhism in these countries was accompanied by strong influences on musical cultures. For instance, the Khotan Buddhist Song (*yutian foqu* 于闐佛曲) found in the Music of Western Liang (*Xiliang yue* 西涼樂), which is part of the Nine-Part Music repertoire, is obviously from the tradition of Buddhist originals.[19]

The early-Tang reorganization of Buddhist musical and dancing offerings, which projected imperial imagery into the monastic courtyard, is broadly captured in the surviving visual materials. In Dunhuang and many other cave temple sites, the portrayal of sophisticated, well-organized performances of music and dance first appeared in the early Tang period and is observed in the dharma preaching scenes of illustrations depicting the Buddhist Pure Land (Z. Wang 2018, 2019). The performance, typically occupying a large space in front of the central preaching Buddha, is depicted with two or four dancers performing on a square platform, accompanied by musicians flanking both sides and playing a variety of instruments (Figure 4). As art historians have presented convincing arguments linking these paradisal performances to Music of Seated-Performers (*zuobuji* 坐部伎), one of two types of entertainment music played at Tang imperial banquets, it is reasonable, therefore, to consider them as visual representations of actual rituals (Zheng 2002, p. 83).[20]

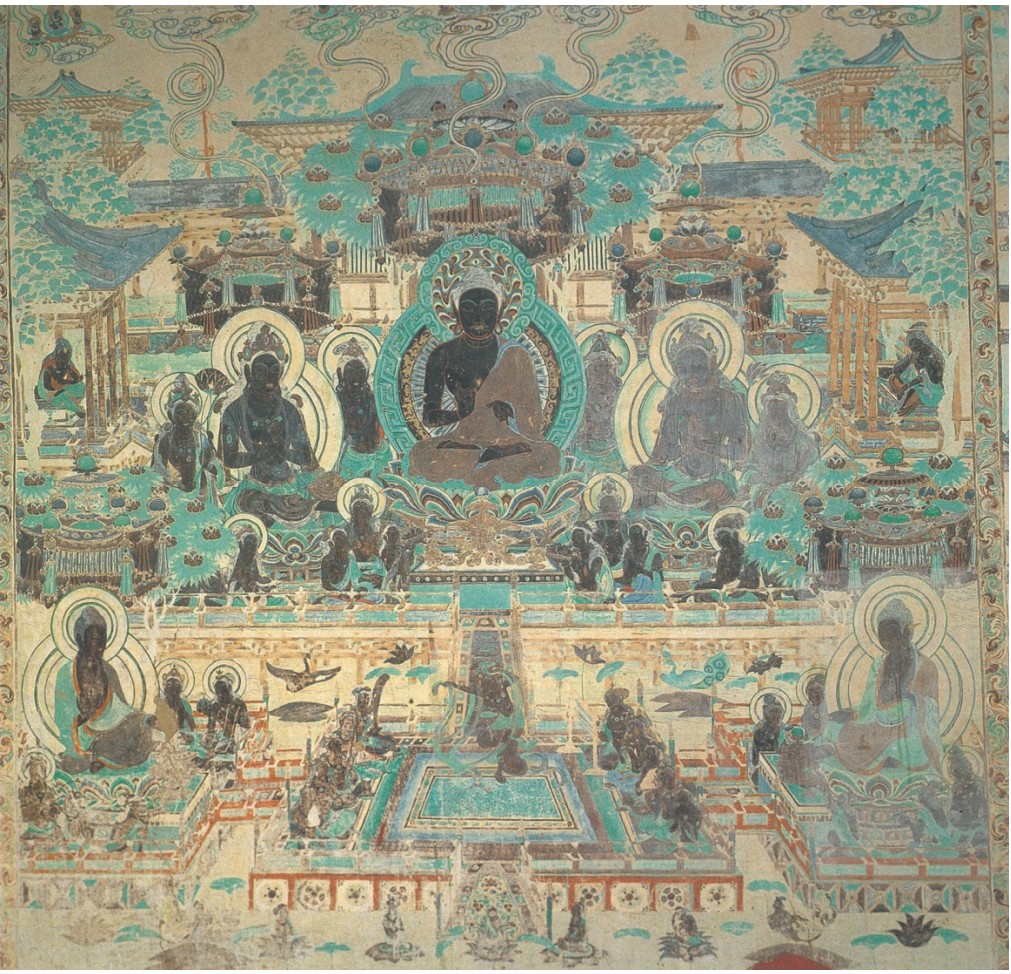

**Figure 4.** Music and Dance Performance in the Dharma Preaching Scene of Pure Land, Northern Wall, No. 320, Mogao Cave, High Tang (Sun and Sun 2001, Figure 108).

The display and performance of the Nine-Part Music in a Buddhist cloister not only simply injected the imagery of imperial authority but, more significantly, introduced a ritual-spatial order that envisioned the corridor as a place for venerating the sacred beings. The details of organizing the Nine-Part Music performance in court banquets were outlined in the early-eighth century state ritual code *Datang Kaiyuanli* 大唐開元禮, which mentions it as an alternative repertoire played during the New Year's Day Banquet (*yuanhui* 元會) (Xiao 2000, pp. 452–56). The banquet, as the second event of the state New Year celebration, followed the morning grand audience held at the imperial palace's main audience compound (Figure 5).[21] Ritual music performed for the audience was provided by a musical ensemble called *gongxuan* 宮懸 (Palace Hanging Instruments), situated in the center of the main audience courtyard. This ensemble consisted of musicians standing before a rack of bronze bells and jade chimes on all four sides, forming a quadripartite band. Except for the emperor in the main audience hall, attendees were seated to the east, west, and south of the ensemble in the courtyard, organized according to their respective ranks and status. As the performance concluded and the banquet began, officials and guests above the third rank from the southern side were permitted to enter the main audience hall, while others maintained their positions. For the banquet music performance, an instruction states that if the court prefers the Nine-Part Music, the *gongxuan* ensemble should be replaced, allowing the Nine-Part musicians and dancers to assume their positions in the courtyard.[22]

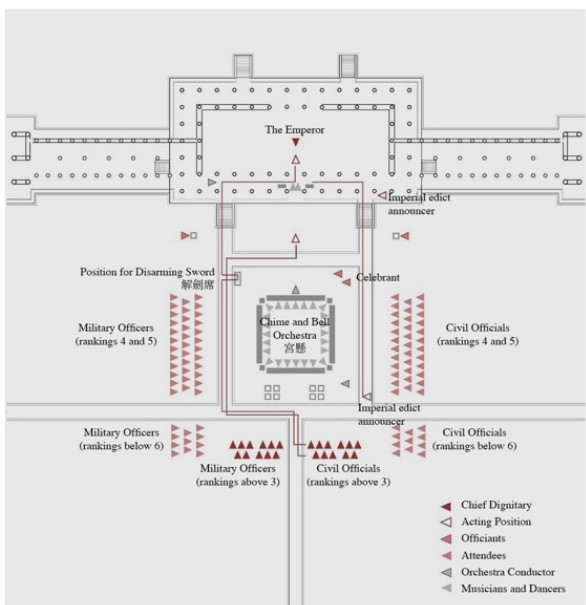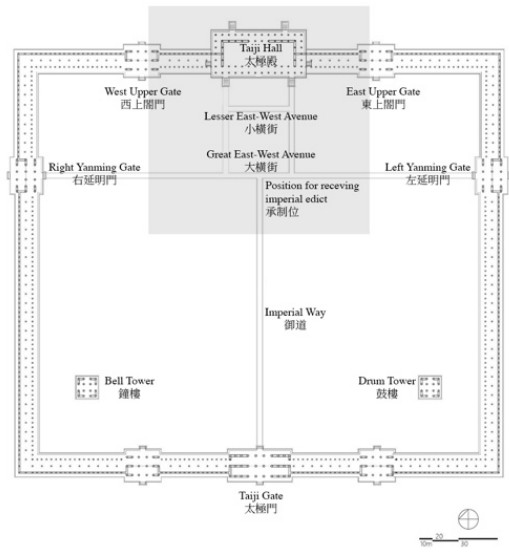

**Figure 5.** Ceremonial Plan of the New Year's Audience in Taiji Palace, based on the Kaiyuan Ritual Code, 7th century (Guo and Shen 2022).

In the ceremonial plan of the New Year's Day Banquet, the central Nine-Part Music ensemble provided musical and dancing entertainment to spectators on three sides of the courtyard: the emperor and high-ranking officials in the northern audience hall, along with the lower-ranking officials in the eastern and western grounds of the courtyard. When adapted into the Buddhist context, particularly for a maigre feast, which could be perceived as the monastic version of an imperial ceremonial banquet, the performance would similarly evoke the same ritual structure of a three-sided offering, envisioning the corridor as a space, albeit of lesser importance, comparable to the image hall for venerating Buddhist pantheons (Figure 6). This observation brings us to another intriguing point: the first recorded performance of the Nine-Part Music in Buddhist celebration in 648 likely coincided with its introduction into the

imperial Near Year banquet, which suggests a possible exchange and influence between the ceremonial plans of both court and monastic banquets.[23]

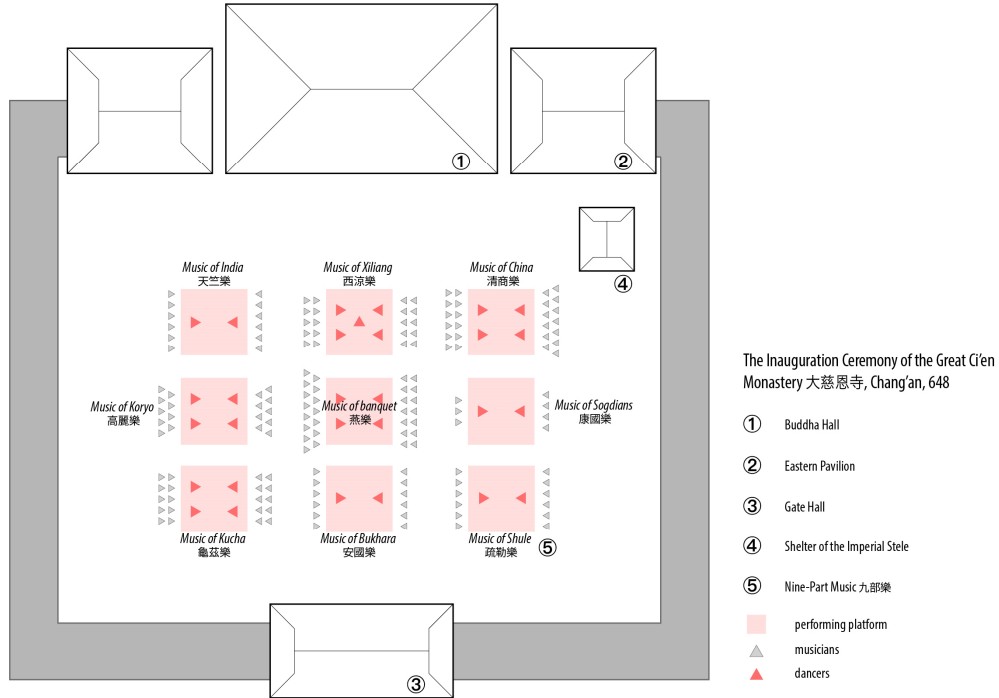

**Figure 6.** The Inauguration Ceremony of Great Ci'en Monastery at 648. The exact number of musicians and dancers for each band of Nine-Part Music is provided in (Zuo 2010, pp. 93–98). However, as no evidence suggests the spatial order of the nine bands, the arrangement presented here is merely the author's hypothetical reconstruction. In this reconstruction, the first band, *yanyue* 燕樂, is positioned in the center because the Dance of Triumph (*pozheng wu*), which is a part of the *yanyue* repertory, was actually performed in the 648 inauguration celebration. The other bands are arranged enumeratively, starting from the northeastern corner and proceeding counter-clockwise.

While the discussions of maigre feast, divine monk veneration, and courtyard entertainment collectively provide many plausible clues towards the consecration of corridor space and the development of corridor paintings, we must not completely disregard the potential impact from abroad, given the highly international nature of the Sui and Tang cultures. The *Mūlasarvāstivāda-vinaya*, translated by Yijing 義淨 (635–713) in the early eighth century, gives a lengthy passage concerning the thoughtful design of pictorial programs at the famous Jetavana vihāra, in which the Buddha not only lists the appropriate paintings but also indicates where each kind of image is to be placed (Wut 2020). *Jātaka* stories of sacrifice offerings, body-sacrifice, or tolerance of mistreatment, he says, should be painted in the verandahs (Yijing 1924–1933a, 656b23–25; 1924–1933b, 283b4). The verandah, serving as the passageway that runs along the front of monastic cells, which, in turn, surround the quadrangular court of the vihāra, is a structure akin to the Chinese Buddhist corridor. The most vivid examples of such painted vihāra verandahs are seen in the two fifth-century Ajanta Caves (No. 1 and 2), which represent actual practices then popular in India. Drawing a direct connection between the Indian vihāra verandah and the Chinese Buddhist corridor is undoubtedly problematic due to the different functions they served. However, the idea of using wall paintings to suggest and reinforce the didactic, cultic, and ritual functions of specific spaces may have been introduced to China by pilgrims and missionary monks. For instance, *Xiuchan Yaojue* 修禪要訣 (Essentials of Cultivating Meditation), a meditation manual produced no later than the ninth century, records a recommendation made by the seventh-century Indian monk Buddhapālita 佛陀波利 that, to protect meditative practitioners from the demon Māra's disturbances, por-

traits of both divine and ordinary monks in seated meditative postures should be painted on the interior walls of meditation halls (Yuan and Li 2020).

## 3. Corridor in Constructing Patriarchal Lineages

Iconographically, the subject of divine monks that became prevalent during the seventh and eighth centuries was often presented as various groupings of patriarchs who protected the transmission of Buddhist teachings. Early groupings in scriptural sources include especially the list of twenty-four patriarchs given in the fifth- or sixth-century Tradition of the Causes and Conditions of the Dharma-Treasury Transmission (*Fu fazang yinyuan zhuan* 付法藏因緣傳) (Kivkara and Tanyao 1924–1933), and the famous sixteen great arhats given in the mid-seventh century Record of the Abiding of the Dharma spoken by the Great Arhat Nandimitra (*Skt*. Nandimitrāvadāna, *Da aluohan Nantimiduoluo suoshuo fazhu ji* 大阿羅漢難提蜜多羅所説法住記) (Xuanzang 1924–1933). This period also witnessed an increasing emphasis on linking sectarian patriarchs genealogically with the Buddha, leading scholars to trace parallel developments in visual traditions (Wong 2018, pp. 154–61). Nonetheless, given that pictorial materials only exist in a handful of cave chapels with unclear sectarian affiliations, scholarly proposals connecting them to particular schools or movements remain tentative (Wong 2018, pp. 159–60). At the level of textual description, most of the available information is associated with monk portraits adorning the corridor walls of urban monasteries. Unlike the cramped space of a cave chapel, the long and continuous corridor wall is by nature ideal to display the master-disciple sequence of dharma transmission. Although rarely did one single document take care to lay out the entire pictorial program, it is our fortune that they occasionally did. The stele inscription that records the reconstruction of Dayun Monastery 大雲寺 at Liangzhou 涼州 (present-day Wuwei city 武威) in 711, for example, provides a glimpse of the full pictorial program of the encircling corridor in the monastery's newly built subsidiary cloister, which consists of the "arhats and divine monks who transmitted the dharma 付法藏羅漢聖僧變", the "Kāśyapa Mātaṇga and Dharmaratna's introduction of dharma to China 摩騰法蘭東來變", and the "avadāna tale of the seven maidens 七女變."[24] The conjoining of the Indian Buddhist saints and patriarchs with the two missionaries who marked the beginning of Chinese Buddhism clearly represents a non- or proto-sectarian approach to bridge the Sino-Indian dharmic divide. This may further suggest the development of a well-established visual tradition by the eighth century, which was seen even at the northwestern frontier. Of course, monasteries in capital cities, which were close to vital centers of various Buddhist schools and practices, were more likely the place where novel ideas could flourish. The following examination will focus on the diverse range of pictorial programs found in corridor wall paintings from several urban headquarters in Tiantai, Chan, and some other schools or movements. This analysis aims to enrich our knowledge of the Tang visual construct of sectarian patriarchal lineages.

### 3.1. Patriarchs for Buddhism Entering China: Ximing Monastery at Chang'an

Ximing Monastery, built by Emperor Gaozong 高宗 (r. 649–683) in the mid-seventh century in Chang'an, stands as an example of a proto-sectarian construct of corridor wall painting. In *Lidai minghua ji*, epigraphs on the east corridor of Ximing Monastery are noted (no. 10, Table A1). The east corridor likely refers to the corridor situated on the eastern half of the monastery's central cloister, typically consisting of three sections: the northern part, which is connected to the image hall; the southern part, which is connected to the gate hall; and the intermediate eastern part, which lies between the northern and southern sections. The eastern section, numbered north-to-south, features distinct epigraphs in the first, third, and fourth bays: *Chuanfazhe tuzan* 傳法者圖贊 (Eulogy on the painting of patriarchs who transmit the Dharma) by Chu Suiliang 褚遂良 (596–658) in the first bay, and eulogies of Lifang 利防 and Dharmakāla 曇柯迦羅 by Ouyang Tong 歐陽通 (625–691) in the third and fourth bays, respectively.[25] Despite the cursory and sporadic nature of Zhang Yanyuan's notes, driven by his interest in great calligraphy masters, the information provided is valuable as it belongs to a larger pictorial program depicting a group of patriarchs under the title of *Chuanfazhe*, the

Transmitters of the Dharma. With the *Chuanfazhe tuzan* prefacing the program in the first bay, the following bays presented portraits of patriarchs in chronological order, each of whom was deemed an important contributor to the transmission of Chinese Buddhist teachings. Li-fang, described in Sui-Tang apocryphal tales as delivering sūtras to the First Emperor of Qin 秦始皇 (r. 221–208 BC), and Dharmakāla, known for translating China's first *vinaya* text in the third century, for example, were both Indian missionary priests traveling to China for the transmission or translation of Buddhist scriptures (Zürcher 2007, pp. 19–20, 55–56).

Notably, Zhang Yanyuan was not the sole individual who ever documented the paintings of Ximing Monastery. Daoxuan 道宣 (596–667), a renowned scholar monk, was appointed the first head monk 上座 ("top seat") of the Ximing Monastery between 658 and 664 (Fujiyoshi 2002, pp. 150–55). During his tenure, he compiled multiple records for the monastery, preserving its splendor and accomplishments. These works are found in Tang and Song catalogs, including the inventory-like *Ximingsi lu* (Record of Ximing Monastery 西明寺錄) in 659, and two textual collections of pictorial inscriptions in 660, namely, *Shengji jianzai tuzan* 聖跡見在圖贊 (Eulogy on the paintings of existing sacred sites and monuments) and *Fohua dongjian tuzan* 佛化東漸圖贊 (Eulogy on the paintings of Buddhist acculturation from India to China) (Yuanzhao 1924–1933, 764c28; 1975–1989b, 650c5–6 and a2–8; Daoxuan 1924–1933a, 282a23–24). Although the two *tuzan* appeared to be no longer known by monks in the 11th century, a recent investigation of the fragments of the Tang manuscript *Huatu zanwen* 畫圖讚文 (Eulogies for Painted Images) provided a significant clue (Dingyuan 2017). This examination reveals that the fragments, composed by combining tales of miraculous images, pagodas, and monasteries in India and China with texts quoted from *Tonglüe Jingzhuzi Jingxing famen* 統略淨住子淨行法門 (Abridged Methods of Pure Practices of the Pure Abider)–Daoxuan's recompilation of a Southern Qi Buddhist treatise–were part of the *Shengji jianzai tuzan*.

It further suggests that the *Shengji jianzai tuzan* texts were inscriptions gathered from wall paintings of Ximing Monastery, serving a propagational purpose in regulating human moral behavior. From the 9th century or even earlier, the original title of *Shengji jianzai tuzan* had fallen out of use. Instead, it was amalgamated with another unknown work for painting inscriptions of Ximing Monastery under a new title, either *Zhufa tuzan* 住法圖贊 (Eulogies for the Paintings of Who Preserved the Dharma) in (Yuanzhao 1975–1989b, 650a8–9) or *Ximing tuzan* 西明圖讚 (Eulogies on the Paintings of Ximing Monastery) in (Jingxiao 1975–1989, 70d1).

It is not difficult to recognize that the unknown work of painting inscription collection in the *Zhufa tuzan*, which has not been discussed in the existing examination of the *Huatu zanwen* manuscript, should be the *Fohua dongjian tuzan*. This is known by the fact that *Zhufa tuzan* remained complete during the Southern Song dynasty, and its texts were frequently referenced in commentaries on Daoxuan's *vinaya* writings between the 10th and 12th centuries. A search of these excerpts from the *Zhufa tuzan* (occasionally titled *Ximing tuzan* 西明圖贊) reveals a wide range of themes, including:

1. The sixteen arhats: Piṇḍola 賓頭盧, Kanakavatsa 迦諾迦伐蹉, Kanaka Bhāradvāja 迦諾跋梨惰闍, Subinda 蘇頻陀, Nakula 諾矩羅, Bhadra 跋陀羅, Kālika 迦理迦, Vajriputra 伐闍羅弗多羅, Gopaka 戍博迦, Panthaka 半托迦, Rāhula 羅怙羅, Nāgasena 那伽犀那, Aṅgaja 因揭陀, Vanavāsin 伐那婆斯, Ajita 阿氏多, and Kṣudrapanthaka 注荼半托迦—*Lüzong xinxue minju* 律宗新學名句, compiled in 1094, (Weixian 1975–1989, 695b14–18);

2. The lineage of the twenty-five patriarchs: the Buddha as the founding teacher and the succession of twenty-four masters from Kāśyapa 迦葉 to Siṃha 師子 who transmitted the teachings—*Sifenlü xingshichao zichiji* 四分律行事鈔資持記, compiled in 1078–1116, (Yuanzhao 1975–1989a, 161a9–11);

3. The arrangement of three high seats in a cave for the first Buddhist council: one for Kāśyapa who presided over the council, one for Ānanda 阿難 and Upali 優波離 who recited the Buddha's teachings, and one for the scriptures transcribed on palm leaves following an unanimous decision among the council members—*Sifenlü xingshichao jianzhengji* 四分律行事鈔簡正記, compiled in the early 10th century (Jingxiao 1975–1989, 13c20–21);

4.  Ānanda's encounter with young ladies and the formulation of a monastic dress code—*Yibo mingyizhang* 衣鉢名義章, compiled between 1042–61, (Yunkan 1975–1989, 601a9–13);

5.  A quote from the *Mahāparinirvāṇa* introducing people to the four stages of awakening: sotāpanna, sakadāgāmi, anāgāmi, and arhat—*Shimen guijingyi tongzhenji* 釋門歸敬儀通真記, compiled in the first half of the 12th century, (Liaoran 1975–1989, 485a22–b2);

6.  The shape of the Jambudvīpa continent—*Sifenlü xingshichao jianzhengji*, (Jingxiao 1975–1989, 127b5–7);

7.  Explanation of the Buddha, the Dharma, and the Sangha—*Shimen guijingyi tongzhenji*, (Liaoran 1975–1989, 462b15–23);

8.  Several significant anti-Buddhist and pro-Buddhist events from the historical periods of the Great Xia (407–431), Northern Wei (386–535), and Northern Zhou (557–581) dynasties—*Shimen guijingyi hufaji* 釋門歸敬儀護法記, compiled in 1150, X59, (Yanqi 1975–1989, 446b1–12).

Excerpts 7 and 8, as identified in the previous examination, are sourced from *Shengji jianzai tuzan*. It is likely that the geographic information of Jambudvīpa (excerpt 6) also comes from the same text, given its strong association with sacred sites. However, the subjects of the sixteen arhats (excerpt 1), the twenty-four patriarchs (excerpt 2), and those pertaining to the deeds of these patriarchs and enlightened beings (excerpts 3, 4, and 5), are distinct in their close connection to the theme of Indian sainthood in the protection and transmission of Buddhist teachings. This immediately brings to mind Daoxuan's *Fohua dongjian tuzan*, as the title itself implies a related subject matter. Additionally, it is worth noting that *Fohua dongjian tuzan* is the only other painting inscription work compiled by Daoxuan in conjunction with *Shengji jianzai tuzan* in the year 660.

Despite the absence of concrete evidence, it is plausible that the *Fohua dongjian tuzan* documents corridor paintings at Ximing Monastery, mainly because *Chufazhe tu*, the pictorial program in the east corridor depicting Indian missionaries in China for dharma transmission, aligns perfectly with the subject matter of *Fohua dongjian*. This enables us to reconstruct a comprehensive program: the western corridor features the sixteen arhats as the dharma-protector and the twenty-four patriarchs as the transmitters of dharma in India proper; meanwhile, the eastern corridor showcases a group of Indian missionaries responsible for transmitting the dharma from India to China (Figure 7). Jointly, these patriarch portraits constructed a narrative of dharma transmission that bridged the Sino-Indian divide, potentially serving as an influential model throughout China, as demonstrated by the earlier mentioned Dayun Monastery example.

From its founding in 658 until the early reign of Xuanzong 玄宗 (r. 712–756), Ximing Monastery was indisputably the epicenter of Buddhist academic teachings, attracting a number of foreign pilgrims who traveled to China to study Buddhism. The significance of Ximing Monastery extended to Japan in the Tenpyō period (729–749), as Daianji 大安寺, the state monastery at Heijōkyō 平城京, was reportedly modeled after Ximing Monastery while it was reconstructed between 729 and 742 under the supervision of Dōji 道慈, a Japanese pilgrim monk studying at Chang'an between 702 and 718 (Wong 2018, pp. 154–61). Although modern scholars remain cautious about the extent to which Dōji's reconstruction followed the Chinese model, recent studies have highlighted the double-sided corridor and the arhat paintings in Daianji's central cloister, suggesting that these novel art and architectural ideas were brought back from the continent (Uehara 2021; Wong 2018, pp. 154–61). The record of arhat portraits, from the Daianji inventory of its property assets compiled in 747, lists ninety-four images of *luohan*, completed in 736, in the monastery's eastern and western corridors (Ōta 1977, p. 53).

Considering Daoxuan's collection and compilation of pictorial inscriptions from Ximing Monastery, I aim to expand upon the established Ximingsi-Daianji connection by suggesting that their corridor paintings not only have a common theme of divine monks but also likely adopt similar iconographic content of "patriarchs for Buddhism entering China." Unlike later visual and textual materials claimed to be Dōji's records, Daoxuan's work on the pictorial inscriptions of Ximing Monastery gained recognition in Japan no later than the ninth century. This is known from the text titled *Ximingsi tuzan* 西明寺圖贊 found in a bibliography compiled

in 891 for documenting the imperial collection of books (Fujiwara 1966, p. 44). The manuscript has been recognized as being the same as the aforementioned *Zhufa tuzan* or *Ximing tuzan*, comprising both the *Fohua dongjian tuzan* and the *Shengji jianzai tuzan* (Dingyuan 2017). The historical details of the manuscript's return to Japan are unclear, however; Dōji could be a plausible candidate, considering his exposure to Daoxuan's scholarship during his time in Chang'an (Wong 2018, p. 161). Moreover, the manuscript, which is not found in other pilgrim monks' catalogs, could have been accessible for Dōji to copy from the rich collection housed in the repository of Ximing Monastery. Taking into account the images of ninety-four arhats in the Daiaiji inventory and the established Ximingsi-Daianji connection, it is reasonable to speculate that there were fifty-four *Chuanfazhe* figures standing in the east corridor of Ximing Monastery's central cloister.

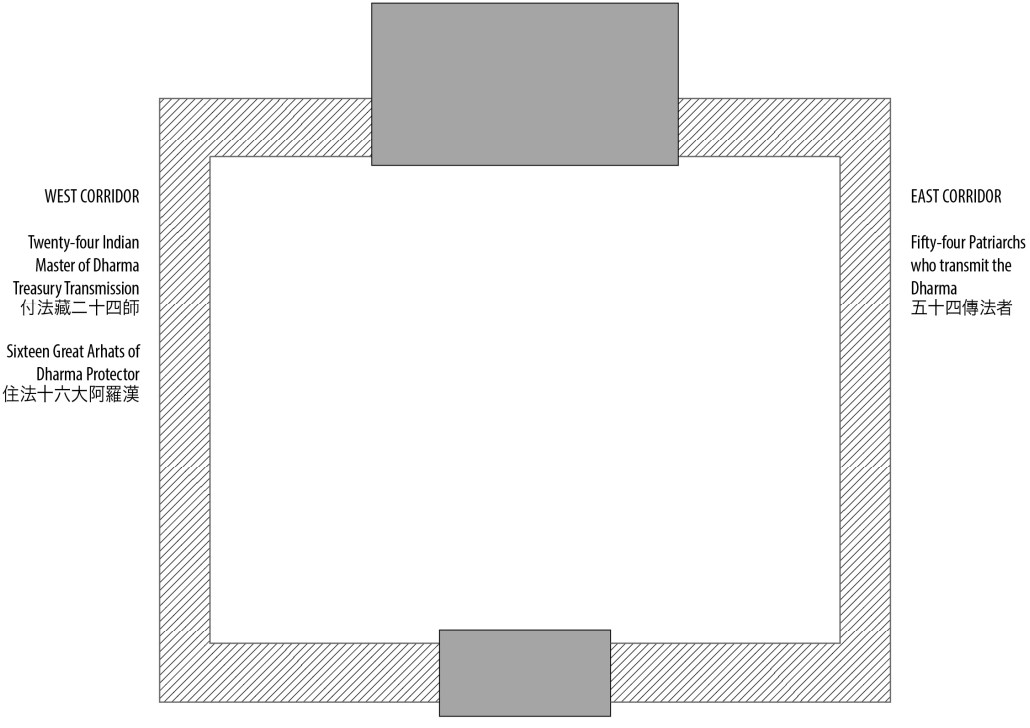

WEST CORRIDOR

Twenty-four Indian
Master of Dharma
Treasury Transmission
付法藏二十四師

Sixteen Great Arhats of
Dharma Protector
住法十六大阿羅漢

EAST CORRIDOR

Fifty-four Patriarchs
who transmit the
Dharma
五十四傳法者

Central Cloister, Ximing Monaster 西明寺, Chang'an, 658

**Figure 7.** A theoretical illustration of the pictorial program of the central cloister's corridors, Ximing Monastery, Chang'an (author's reconstruction).

Regarding the origin of the *Fohua dongjian tu*, although Daoxuan is recognized as the compiler of the pictorial inscription document, which serves as the primary source for our information about the corridor paintings, the actual authorship should be attributed to another individual. The precise date of the paintings remains unknown, but they must have been created no later than 655. This is because Chu Suiliang, the calligrapher responsible for *Chuanfazhe tuzan*, was demoted by Emperor Gaozong and sent away from Chang'an in the late autumn of that year. He never returned to the capital until his death in 658.[26] Hence, the creation of the paintings fell within the period between 652, when the monastery was initially established by converting a deceased prince's residence, and 656, when Emperor Gaozong bestowed the title of Ximing upon the monastery and initiated the grand reconstruction (Zhanru 2022, pp. 53–61). This period coincided with the early reign of Emperor Gaozong, during which Xuanzang, the *de facto* leader of Chang'an Buddhism, enjoyed great honor from the imperial family. It is highly plausible that Xuanzang was the individual responsible for the work, particularly given that a few years later, the emperor appointed him to inspect the monastery site before the commencement of the reconstruction. In contrast, Daoxuan, who joined Xuanzang's translation team between 646 and 658, did not actively engage

with the imperial religious projects until he was appointed as the head monk of the Ximing Monastery in 658 (Fujiyoshi 2002, pp. 145–62). A comparable strategy in designing iconography programs can be seen decades later by one of Xuanzang's close disciples, which may provide further evidence for the hypothesis on authorship. After Xuanzang's death in 664, Jingmai 靖邁, who followed Xuanzang for twenty years, commissioned paintings on the interior walls of the translation hall in the Great Ci'en Monastery to honor his master. These paintings featured a group of one hundred and twelve translators, ranging chronologically from Kāśyapa Mātaṅga to Xuanzang (Liu 2017). While Jingmai's work differed from the corridor pictorial program, as it included not only monastic but also layman translators, they shared similar concepts.

The corridor paintings at Ximing Monastery hold a significant place in the history of Chinese Buddhist art. Firstly, as the earliest visual representation to bridge the Sino-Indian dharma divide, these paintings diverge from the typical sectarian approach of connecting genealogies of Indian and Chinese masters, which was first seen in the writings of Tiantai luminary Guanding 灌頂 (561–632) half a century earlier (Young 2015, pp. 67–69). Distinctively, the corridor paintings sought to embody the subject not by forming master-disciple lineages but rather by embracing a diverse group of prominent monastic missionaries and translators, who were well known by the Chinese through the Sui and early-Tang historiographies of Buddhism. Secondly, the presence of the sixteen arhats is notable, as it is the first known attempt to visualize this novel subject in China following the translation of Nandimitrāvadāna in 654, and was conceived by Xuanzang, the translator himself. This discovery sheds new light on the scholarly investigation of the early development of *luohan* iconography in China.[27]

### 3.2. The Forty-Two Xiansheng and Monks Copying-Reciting Lotus Sūtra: Tiantai Corridor Paintings

The Tiantai School, as one of the earliest local Chinese Buddhist traditions to form a coherent and distinct movement, witnessed its advocates undertaking the unprecedented endeavor to establish genealogical connections between Indian and Chinese masters (Young 2015, pp. 125–30). This sectarian effort was represented in the form of visual arts along the Buddhist corridor, exemplified by the Forty-two *Xiansheng* 四十二賢聖 images painted in the Baoying Guanyin Cloister 寶應觀音院 of Zisheng Monastery 資聖寺 at Chang'an (no. 18, Table A1). In a previous scholarly study, the Forty-two *Xiansheng* at Zisheng Monastery was identified as representing the forty-two stages of the bodhisattva path, as enumerated in the fifth-century *Pusa Yingluo Benye Jing* 菩薩瓔珞本業經 (Sūtra of the Diadem of the Primary Activities of the Bodhisattvas). These stages signify the spiritual progression from the initial aspiration to the ultimate attainment of Buddhahood (Liu 2013). However, this identification is mistaken, as *Youyang zazu* gives a clear account of Nāgārjuna 龍樹 and Śāṇavāsa 商那和修 as members of the Forty-two *Xiansheng* (Duan 2015, p. 1925), which serves as compelling evidence that the Tiantai pictorial program was not concerned with personifying Bodhisattva stages. The inclusion of Nāgārjuna and Śāṇavāsa suggests that the twenty-four patriarchs from the Dharma-Treasury Transmission could be part of the Forty-two *Xiansheng* program, as it is the only known grouping containing both Indian masters. In addition, Ennin's catalog submission to the Japanese court reveals that during his 840–845 stay at Zisheng Monastery, he reproduced mural texts and compiled them into a single-volume manuscript titled "Eulogies from Portrait Paintings of Nanyue, Tiantai, and others on the walls of Baoyin Guanyin Cloister at Zisheng Monastery 長安資聖寺寶應觀音院壁上南岳天台等真影讚 (Ennin 1924–1933, 1084a23–25)." This strongly implies the presence of Master Huisi of Nanyue 南岳慧思 and Master Zhiyi of Tiantai 天台智顗, who are regarded as the key founding patriarchs of the Tiantai School.[28]

The inclusion of Huisi and Zhiyi immediately draws our attention to Tiantai master Jingxi Zhanran's (711–782) 荊溪湛然 ritual work *Qing Sishier Xiansheng Yi* 請四十二賢聖儀 (Rites of Inviting the Forty-two Saints).[29] Zhanran, accredited with his substantial contributions to the revival of the Tiantai tradition, held a pivotal role in the formation of the sectarian ideology (Tang 2008, pp. 132–35). Zhanran's influence on the Tiantai lineage extends

beyond providing a list of five successive patriarchs as lineal descendants of Zhiyi. He also re-emphasized the link of Zhiyi, Huisi, and Huiwen 慧文 (active in the 550s) with the Dharma Treasury Transmission lineage, first claimed by Guanding in his introduction to the *Mohe zhiguan* 摩訶止觀 (The Great Calming and Contemplation) (Young 2015, p. 135; Lin 2006). This connection is not established through direct master-disciple transmission but through Nāgārjuna's *Da Zhidu Lun* 大智度論 (Great Perfection of Wisdom Treatise) (Zhiyi 1924–1933, 1a13–b8). By grouping the Indian monastic divinities with the three masters of latter-day China, the making of the Forty-two *Xiansheng* and its offering ritual were doubtless an effort to propagate the superiority of Tiantai traditions, as demonstrated by Ennin's account of witnessing the performance of the Forty-two Xiansheng offering at a monastery in Yangzhou (Ennin 2007, pp. 97–98).

Aside from the Indian dharma-transmission patriarchs and Chinese Tiantai masters, the remaining figures could number either fifteen or sixteen.[30] There is limited information about these unidentified beings; however, they are unlikely to be associated with the five Tiantai masters after Zhiyi due to the numerical discrepancy. Nonetheless, the number sixteen brings to mind the well-known grouping of the sixteen great arhats, especially when considering the established practice of combining them with the Dharma-Treasury Transmission lineage, as seen in Ximing Monastery, which was located alongside Zisheng Monastery in the same Chang'an city. Should this hypothesis prove valid, the Forty-two *Xiansheng* might intriguingly represent a Tiantai adaptation of Daoxuan's *Fohua dongjian tu*, thereby forming a pictorial program characterized as—borrowing the phrase from the 711 Dayun Monastery stele inscription—"arhats and divine monks who transmitted the dharma".

A further examination of the Guanyin Cloister in Zisheng Monastery, where the portraits of the Forty-two *Xiansheng* were painted, may uncover its previously unexplored connection to the Tiantai tradition. Textual materials, though somewhat ambiguous, suggest that the Guanyin Cloister may have had a circular pagoda at its courtyard center, leading to its alternative name, Circular Pagoda Cloister (*yuantayuan* 圓塔院).[31] In Duan Chengshi's record of Zisheng Monastery, both Guanyin Cloister and Circular Pagoda Cloister are present, which would typically refer to two separate compounds. However, the description of the Guanyin Cloister and its corridor paintings by forty-two *Xiansheng* is uncharacteristically embedded within the paragraph discussing the Circular Pagoda Cloister. Moreover, it is notable that the north hall of the Circular Pagoda Cloister, likely its main image shrine due to the location, housed a three-*zhang* (approximately nine-meter) tall iron Guanyin statue (Duan 2015, p. 1925). The colossal image of Guanyin, being the principal focus of devotion alongside the circular pagoda within the same cloister, also brings us to mind the name of the Guanyin Cloister.

Exploring Zisheng Monastery's history and its link to Tiantai practice would be helpful to clarify the dual naming phenomenon. Originally founded in 663, the monastery suffered a devastating fire in 703 but was promptly reconstructed (Ono 1989, p. 67). Following the restoration, it rose to prominence as a notable center for Buddhist teachings in Chang'an until the mid-ninth century when anti-Buddhist persecution occurred. The construction date of the Guanyin Cloister is not specified in any available sources. However, given that the artists responsible for the circular pagoda and corridor paintings were active during the reigns of Emperor Daizong 代宗 (r. 762–779) and Dezong 德宗 (r. 779–805), the compound was likely either built or underwent renovations during that period.[32] The time frame is concurrent with the Tiantai monks' activities at Zisheng Monastery. Daguang (736–805) 大光, renowned for his Lotus Sūtra chanting, was appointed abbot of both Qianfu Monastery 千福寺—the Tiantai headquarter in Chang'an—and Zisheng Monastery by Emperor Suzong 肅宗 between 761 and 762 (W. Xu 2003). Later, Tiantai scholar-monk Daoye 道液, who stayed in Zisheng Monastery during the reigns of Emperors Daizong and Dezong, Daoye, became influential in Japan for his exegetical writings (Sato 2013). The confluence of active Tiantai monks in Zisheng Monastery and the production of artworks in the Guanyin Cloister during the latter half of the eighth century provides compelling evidence to support the hypothesis that the cloister experienced a significant renovation during this period. which, potentially including

the erection of a circular pagoda and the repainting of corridors, was likely facilitated by the monastery's Tiantai community, leading to the new name "Circular Pagoda Cloister."

The iconographic representation of the Forty-two *Xiansheng* at Zisheng Monastery is arguably the earliest endeavor to employ a Buddhist corridor for celebrating the sectarian preeminence of Tiantai tradition. This presumption becomes apparent when compared with the Western Pagoda Cloister 西塔院 (also known as the Lotus Cloister 法華院 or Lotus Ritual Arena 法華道場) of Qianfu Monastery, the Tiantai headquarters in Chang'an. Built by Tiantai master Chujin 楚金 (698–759) between 742 and 745, with the generous patronage of Emperor Xuanzong 玄宗 (r. 712–756), the Western Pagoda Cloister was a corridor-enclosed compound featuring a portrait hall in the northern end and a Prabhutaratna Pagoda 多寶塔 at the courtyard center.[33] The Prabhutaratna Pagoda was a sacred structure embodying the teachings of the Lotus Sūtra, and Chujin enshrined thousands of śarīra relics, his self-portrait engraved on a stone, a thousand copies of the Lotus Sūtra, and thirty-six golden-lettered copies of the same scripture under the ground of its foundation. In the latter half of the eighth century, Chujin's Prabhutaratna Pagoda became an influential model for many Tiantai monasteries at Chang'an to emulate.[34] The Circular Pagoda in Zisheng Monastery, which adopted the similar practice of burying a thousand copies of the Lotus Sūtra, stood out as a notable example (Duan 2015, p. 1925). The close connection between the two Prabhutaratna Pagodas does not necessarily suggest a complete replication of the entire compound, as the veneration programs within their main halls and corridors exhibit significant differences. The north hall of the West Pagoda Cloister served as a memorial space for worshipping Huisi, Zhiyi, and the other seven unidentified Tiantai patriarchs and their disciples.[35] In contrast, the north hall of the Circular Pagoda Cloister, likely predating the compound's conversion for Tiantai purposes, functioned as a conventional image hall housing a giant Guanyin statue. Regarding the corridors, the Western Pagoda Cloister, although not fully recorded, displays a collection of loosely connected images with diverse themes, such as a portrait of *Tianshi* 天師 (Celestial Master), a portrait of Chujin, and a scene of Maitreya's descent to this world 彌勒下生變 (no. 14, Table A1).[36] Conversely, the Forty-two *Xiansheng* in the Circular Pagoda Cloister showcases a careful design, wherein the entire corridor space was taken into consideration. Considering that Zhanran, the author of the Forty-two *Xiansheng*, immersed himself in the study of Tiantai teachings in Zhejiang several decades after the construction of the Western Pagoda Cloister (Chen 1999), it appears that the Tiantai sectarian insight given to the corridor's visual significance remained undeveloped by the mid-eighth century.

Another Tiantai practice of grouping, not for patriarchal lineage but rather for the subject of miraculous responses (*ganyin* 感應), is observed in the ninth-century Longxing Monastery 龍興寺 in Yangzhou (no. 22, Table A1). Within the monastery, a compound known as Lotus Cloister 法花院 or Lotus Ritual Arena 法花道場 was dedicated to the practice of Tiantai School. Ennin, during his stay in Yangzhou, visited the cloister and recorded in his diary that the south corridor of the cloister housed portraits of Master Huisi, Master Zhiyi, and over twenty monks who "received miraculous responses by hand-copying and reciting the Lotus Sūtra (Ennin 2007, pp. 90–91)."[37] Although the multitude of paintings proved too numerous for Ennin to fully replicate, he successfully copied portraits of the two masters and created ten sketch drawings from the selection of over twenty monks. The titles of these sketches are preserved in Ennin's catalog (Table 1).[38] Even a cursory examination could show that these monks were featured in Tang-dynasty tales promoting the belief of the Lotus Sūtra as the central scripture of the Tiantai teachings. The early eighth-century *Hongzan Fahua Zhuan* 弘贊法華傳 (Accounts of Glorifying the Lotus Sūtra), arguably the first collection of its kind, contains short biographies of these monks, detailing their experiences with various miraculous responses received as meditators, memorized chanters, intonated reciters, and hand-copiers of the Lotus Sūtra (Li 2014). Apart from the ten figures documented by Ennin, the *Hongzan Fahua Zhuan* includes thirty-nine monastic priests whose stories align with the criteria of "hand-copying and reciting the Sūtra." As a result, they are likely the subjects of the over ten unrecorded portraits painted on the corridor of the Lotus Cloister. Han Gan (706–783) 韓幹, who created the two masters' portraits, was erroneously attributed by Ennin as a painter from the Liang period (502–557).

In fact, he was a court artist active in Chang'an during the rule of Xuanzong (r. 712–756). No records exist regarding Han Gan's whereabouts after the Anshi Rebellion. Nevertheless, it is most reasonable to date the corridor paintings to the latter half of the eighth century.

**Table 1.** Textual Materials for the Portrait Paintings of Monks who "received miraculous responses by hand-copying and reciting the Lotus Sūtra" at Longxing Monastery, Yangzhou.

| *Nittō Shin Gu Shōgyō Mokuroku* 入唐新求聖教目録 (Ennin 1924–1933, 1087a27–b10) | *Hongzan Fahua Zhuan* 弘贊法華傳 |
| --- | --- |
| scene of venerable Huisi of Nanyue unearthing relics from his previous life 南岳思大和尚示先生骨影 | Chen-dynasty monk Shi Huisi from Nanyue 陳南岳釋慧思, chapter of meditator (*xiuguan* 修觀), (Huixiang 1924–1933, 21c12–22b16) |
| scene of Tiantai Master receiving a miraculous image 天台大師感得聖像影 | Sui-dynasty monk Shi Zhiyi from Mt. Tiantai 隋天台山釋智顗, chapter of meditator, (Huixiang 1924–1933, 22b17–23a20) |
| scene of dhyāna master Shandeng beholding gold and silver hall by chanting the Lotus Sūtra 山登禪師誦法花感金銀殿影 | Liang-dynasty monk Shi Zhideng from Mt. Lu 梁匡山釋智登, chapter of memorized chanter (*songchi* 誦持), (Huixiang 1924–1933, 30a20–b20) |
| scene of an araṇya bhikkhu beholding Samantabhadra in the air 阿蘭若比丘見空中普賢影 | foreign araṇya bhikkhu 外國蘭若比丘, chapter of intonated reciter (*zhuandu* 轉讀), (Huixiang 1924–1933, 40b25–c5) |
| scene of dhyāna master Ying drawing audience of benevolent deities by chanting the Lotus Sūtra 映禪師誦法花善神来聽經影 | Sui-dynasty monk Shi Sengying from Yongqi Monastery at Jiangyang 隋江陽永齊寺釋僧映, chapter of memorized chanter, (Huixiang 1924–1933, 33b3–12) |
| scene of the deceased dhyāna master Huixiang's auspicious retribution of lotus blossom and spontaneous sūtra recitations in his grave by (the merit of) his chanting of the Lotus Sūtra during lifetime 惠向禪師誦法花滅後墓上生蓮花及墓裏常有誦經聲影 | Sui-dynasty monk Shi Huixiang from Jiangdu county 隋江都縣釋慧向, chapter of memorized chanter, (Huixiang 1924–1933, 32c28–33a12) |
| scene of venerable monk Fahui chanting the Lotus Sūtra before Yama 法惠和上閻王前誦法花影 | Liang-dynasty Ping Fahui beholding a monk in the underworld梁憑法慧冥道見僧, chapter of memorized chanter, (Huixiang 1924–1933, 31a26–b4) |
| scene of dhyāna master Huibing attracting the worship of heavenly beings by chanting the Lotus Sūtra 惠斌禪師誦法花神人来拜影 | Sui-dynasty monk Shi Huibing from Chanju Monastery 隋禪居道場釋慧斌, chapter of memorized chanter, (Huixiang 1924–1933, 33c6–20) |
| scene of dhyāna master Ding receiving offerings from heavenly boy by chanting the Lotus Sūtra 定禪師誦法花天童給事影 | Liang-dynasty monk Shi Sengding from Chanzhong Monastery 梁禪眾寺釋僧定, chapter of memorized chanter, (Huixiang 1924–1933, 30a13–20) |
| scene of dhyāna master Daochao learning the rebirth place of his untimely-dead disciple by chanting the Lotus Sūtra 道超禪師誦法花感二世弟子生處影 | the deceased disciple of Northern Qi monk Shi Daochao's 北齊釋道超故弟子, chapter of hand-copy scriptures (*shuxie* 書寫), (Huixiang 1924–1933, 42c26–43b9) |
| scene of an elderly monk from Qin prefecture instructing a disciple and receiving a dream that unveils the cause from the disciple's previous life 秦郡老僧教弟子感夢示宿因影 | Monastic novice from East Monastery at Qin prefecture 秦郡東寺沙彌, chapter of memorized chanter, (Huixiang 1924–1933, 28c20–29a6) |
| scene of dhyāna master Fahui emitting a radiant light from his mouth and illuminating the room by chanting the Lotus Sūtra 法惠禪師誦法花口放光照室宇影 | Chen-dynasty monk Shi Fahui from Qushui Monastery at Shouchun 陳壽春曲水寺釋法慧, chapter of memorized chanter, (Huixiang 1924–1933, 32b11–15) |

### 3.3. Patriarchs Conferring Monastic Vestments: The Chan Vision of Corridor Paintings

The Chan Buddhist school, which emphasizes direct dharma transmission through masters, has been known to use corridors to promote patriarchal lineage since the eighth or ninth century. This practice is evident in the Dunhuang manuscript of the Platform Sūtra, the earliest extant version of the foundational guide for the Southern Chan school. Within the text, Huineng recounted his pursuit of dharma from Hongren 弘忍 (601–674), the Fifth Patriarch of Chan School, at East Fengmao Mountain 東馮茂山 in Qizhou 蘄州. Hongren, in order to de-

termine a fitting successor to carry on the teachings, proposed a poem contest among disciples. Concurrently, Hongren sought to create a visual testament to the dharma conferral for posterity. To achieve this, he commissioned a court artist to produce two works: an illustration of the Laṅkāvatāra Sūtra 楞伽變 and a portrait of Hongren himself bestowing the dharma transmission robe 五祖大師傳授衣法. These artworks were displayed on a three-bay-wide wall within the south corridor, situated in front of the Grand Master hall (Fahai 1924–1933, 337b17–20, and 337c3–4). Owing to the well-known sectarian ideology of the Platform Sūtra and its rewriting of history, the historical accuracy of Hongren's corridor art project cannot be confirmed without additional sources (Kinugawa 2016). More plausible is to consider this account as an interpretation shaped by practices seen in the mid-to-late Tang Chan monastic institutions. Nevertheless, the presence of Laṅkāvatāra Sūtra illustration may partially mirror a visual tradition of earlier origins, as this scripture, central to the early Chan movement, was later displaced by the Diamond Sūtra in Huineng's teachings.

In the Platform Sūtra narrative, the Grand Master hall served as a site for dharma transmission, where Hongreng imparted secret teachings and transferred his monastic vestment to Huineng (Fahai 1924–1933, 338a14–17). The narrative implying the Grand Master hall as essentially a dharma hall (*fatang* 法堂) is likely a retrospective interpretation of a seventh-century "Chan" monastery from an eighth-ninth century perspective because the architectural practice of "erecting no Buddha hall and only a dharma hall" for Chan monasteries was introduced more than a century later by master Huaihai 怀海 (749–814). This enables us to perceive the cloister where the Grand Master Hall stood, equivalent to the central cloister of a monastery, and to examine the positional significance of the south corridor in the early sectarian construct of patriarchal lineage. The south-facing tradition of Chinese architecture determines that the east and west sides of a cloister are usually more suitable for a continuous pictorial program than the north and south sides, which are interrupted by buildings along the central axis. In a non-sectarian monastery with east and west corridors featuring the sixteen great arhats and twenty-four dharma-treasury transmission masters, the south corridor became ideal for adding portraits of sectarian patriarchs when the monastery was converted to house a specific school. This location, extending from the east and west corridors, allowed for a spatial presentation of the sectarian patriarchs as a continuation of the existing lineage of Indian masters. The visual-spatial conception transcends sectarian boundaries, as demonstrated by the portraits of Tiantai masters and miraculous monks from the south corridor of the Lotus Cloister at Longxing Monastery.

The examination and discussion of iconographic programs sheds light on the diverse and dynamic ways in which the Tang Buddhist corridor was utilized to construct Sino-Indian connections and Chinese Buddhist patriarchal lineages. Naturally, these examples represent only a small fraction of what existed during the Tang dynasty. For instance, the late-Tang Dunhuang manuscript P. 2971 documents twenty-three figures of divine monks painted on the east corridor wall of a monastery, featuring a diverse array of groupings. These include six from the Buddha's ten disciples (Subhūti 須菩提, Pūrṇa 富樓那, Kātyāyana 迦旃延, Aniruddha 阿那律, Upāli 優波离, and Rāhula 羅睺羅), five from the Dharma-Treasury Transmission lineage (Jayata 闍夜多, Vasubandhu 婆修盤陁, Manorhita摩奴羅, Haklenayashas 鶴勒那夜奢, and Āryasimha 師子比丘), the six Chan patriarchs (Bodhidharma 達摩, Huike 惠可, Sengcan 僧璨, Daoxin 道信, Hongren, and Huineng), Indian scholar-monks (Asaṅga 無著 and Vasubandhu 世親), Indian missionaries (Kumārajīva 羅什 and Fotudeng 佛圖澄), and Chinese monks (Liu Sahe 劉薩訶 and Huiyuan 惠遠) (M. Yang 2018). This unique combination is not found in other textual sources, indicating a highly flexible approach to composing the pictorial program of the corridor space. As demonstrated in the earlier discussion, the flexibility in many sectarian programs was achieved by incorporating additional Chinese patriarchs to expand upon the pre-existing paintings of Indian masters, such as the twenty-four dharma-transmission patriarchs, the sixteen arhats, and so on.

In Tang Buddhist monasteries, portrait halls built to honor deceased eminent monks served as a prominent location for murals of patriarch portraits, but these spaces primarily functioned as solemn memorial sites intended for the monastic community (Sharf and Foulk

1993; Yu 2019). In contrast, the corridor was selected as a portrait gallery for sectarian propagation, not only due to its association with divine monks but also because of its publicity, which offered easier access for lay visitors.

## 4. From Walking to Seated: Towards Static Worship and the Closure of Corridor

Regardless of the potential diversity of functional use, a corridor, with its elongated, narrow, and encircling space, is fundamentally a passageway anticipating non-static activities. The Buddhist corridors retained this characteristic even after being consecrated in Sui-Tang ritual reform, offering an unconventional worship space in motion. Beyond housing audience seating and incense processions in grand public assemblies, the corridor was also designated for the practice of *jingxing* 經行 (perambulation) and was ideal for individuals appreciating murals.[39]

Clearly, all these activities were performed through the action of walking.[40] This brings us to an interesting observation, as *xingseng* 行僧 (monks in perambulation), a type of iconography depicting monks in walking posture and the most popular style of divine monk portrait, first emerged in the corridor during the early Tang era, or even earlier. When portraits of divine monks were first produced in the late fifth century, the figures were portrayed in seated positions (Daoshi 1924–1933, 609c9–10). In previous scholarship, the earliest examples of *xingseng* images date to the Zhenguan era (627–650) at the Shengguang Monastery 勝光寺 in Chang'an (Yu 2011; Kim 2020). In the northwest cloister of the monastery, these images adorned the side doors on the south façade of the modest hall (Zhang 2018, p. 84). This particular location is noteworthy because Tang-dynasty image halls often featured a one-bay deep portico at the front, connecting to flanking corridors and forming a four-sided colonnaded quadrangle (Z. Xu 2020). In this context, the front portico can be reasonably considered as part of the corridor system, making the doors of the modest hall, which also served as the back of the portico, equivalent to the walls of the corridor. Another case potentially predating the Shengguang Monastery is found at Zhaojinggong Monastery 趙景公寺. Here, the eastern corridor displays a *xingseng* who "turns his eyes to look at viewers" (no. 2, Table A1). As Zhaojinggong Monastery was built in 583, its *xingseng* image might be a work from the Sui dynasty.

By analyzing the placement of the *xingseng* images, we can collect further evidence to reinforce the theoretical link between the movement-oriented nature of the corridor and the rise of the *xingseng* image. The survey of documented instances reveals that the *xingseng* images were predominantly depicted on the walls of corridors and corridor-equivalent spaces, such as the doors of the image hall and gate hall (Yu 2011). On rare occasions, they were found on the interior walls of image halls, with the earliest examples from Chang'an at the eighth-century Jianfu Monastery 薦福寺 and Dayun Monastery 大雲寺 (Zhang 2018, pp. 72, 93).[41] It is worth noting that the image halls featuring *xingseng* murals on their interior walls were all situated in subsidiary cloisters, which were intended for monastic use rather than public ceremonies. As pragmatic monastic compounds, these cloisters likely prioritized the ritual significance of the interior space, specifically the ambulatory of a hall, where circumambulation, or the practice of walking in a clockwise direction around the central icons, occurred. Zhang Yanyuan's report of the *xingseng* image in the image hall of the Pure Land Cloister 淨土院 at Dayun Monastery accurately captures this subtle distinction, referring to it as *raodianseng* 繞殿僧 or monks circumambulating the hall. In summary, the manifestation of the *xingseng* image within the interior may be contingent upon the diminished prominence of the corridor as a ritual locus.

The critical transformation of portraiture, driven by ritual-spatial shifts, also bears witness in pictorial materials from cave chapels. Visual representations of patriarchs are found in Dazhusheng Cave 大住聖窟 at Baoshan 寶山 (589) and in Leigutai 擂鼓臺 central cave at Longmen 龍門 (690–2), both illustrating the patriarchal lineage of Dharma-Treasury Transmission (Wong 2018, pp. 156–60). The Dazhushengku engraved small figures of patriarchs in seated posture on the inner face of the entrance wall, distributed in horizontal registers. In contrast, the Leigutai central cave assigns a prominent role to the patriarchs, who are dis-

played in human-scale walking poses, covering the lower half of the north, east, and west walls around the cave (Figure 8). With its original main icon in the center, the Leigutai central cave purposefully organized the layout of patriarch figures to model after a contemporary urban monastery, where paintings of walking monks decorated corridor walls around the cloister.

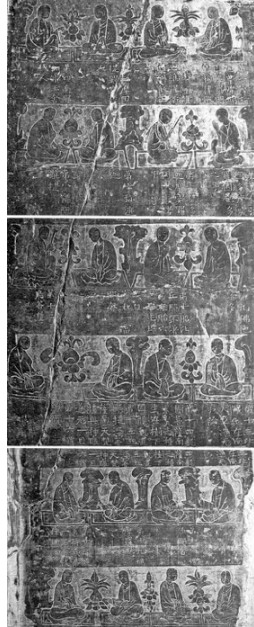 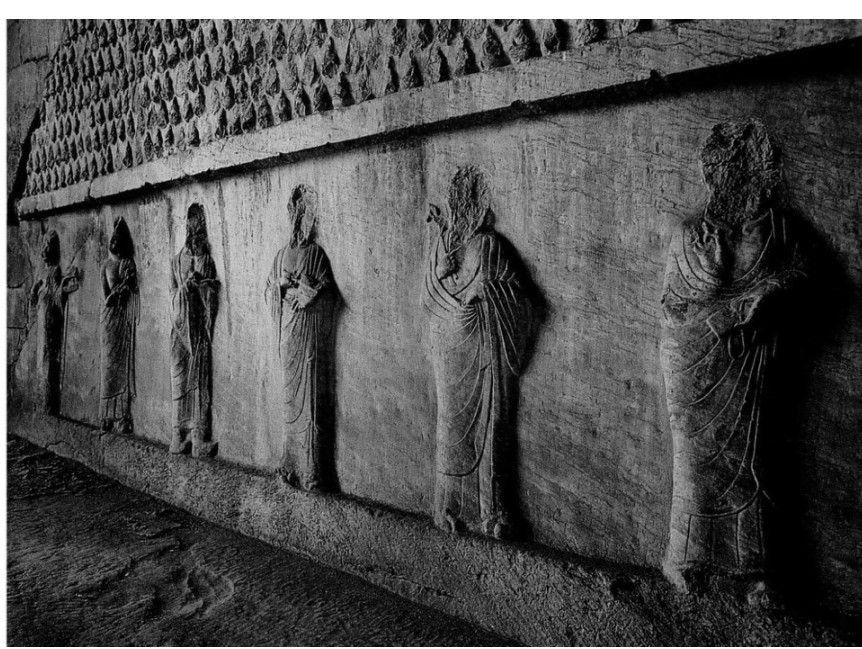

**Figure 8.** Patriarchs who transmit the dharma: left, Dazhushengku 大住聖窟, Baoshan 寶山, Anyang, 589, after (Chen and Ding 1989, pl. 215); right, Leigutai 擂鼓臺 Central Cave, Longmen, 690–2, after (Xia and Sui 1992, pl. 258).

Beyond the monastic portrait, the corridor's mobile nature necessitates that other subjects, although occasionally chosen for display in such an area, exhibit similar characteristics. This was particularly true during the Sui and early Tang periods, when the practice of corridor wall paintings was still a novel concept. The earliest example is a Sui painting of Western Pure Land and the sixteen ways of meditation, undoubtedly illustrating the Visualization Sūtra, from the west corridor of the Sanjie Cloister 三階院 at Zhangjinggong Monastery (no. 2, Table A1). While Zhang Yanyuan's text does not provide more details on the painting's composition, research on Tang Buddhist art indicates that it likely featured a horizontal narrative scene, as exemplified by an early Tang artwork at Mogao Cave 431 (Shi 2002, pp. 89–90). This scene would sequentially depict Prince Ajatasatru's usurpation and the Buddha's discourse to Queen Vaidehi about rebirth in Amitabha's Pure Land. Other early examples, including the early-seventh century hell scenes from Baocha Monastery 寶刹寺 (no. 8, Table A1) and the illustrations of Sūryagarbha and Candragarbha Sūtras 日藏月藏經變, as well as the scenes of various karmic rewards 業報差別變 from Jing'ai Monastery 敬愛寺 (dated to 722, no. 12, Table A1), all fall under the category of narrative paintings.[42] The iconic frontal representation of dharma preaching assemblies, encouraging viewers to stop and contemplate, appeared sporadically in Buddhist corridors by the eighth century. The earliest known example is the illustration of the Diamond Sūtra 金剛經變, painted on the south corridor wall of the Pure Land Cloister 淨土院 in Xingtang Monastery 興唐寺 (dated to 723, no. 16, Table A1).[43]

As the sporadic cases of iconic sūtra illustrations may suggest, the ritual pattern of Buddhist corridors during the middle and late Tang periods experienced a tendency towards immobilization, favoring a more static form of worship. Ennin's account of the monastic celebration of the Lantern Festival in 839 at Kaiyuan Monastery offers a brief glimpse into this transformation. On that night, he recorded: "Monks in the monastery lit lamps and offered them to the Buddha, as well as paying reverence to the portraits of patriarchs. Laymen did

likewise. A lamp tower was erected in front of the Buddha hall. In the courtyard and along the sides of the corridors, they burned oil lamps that were too numerous to count."[44] This text seems to follow an order, first mentioning the enshrined image and then the enshrining space, which suggests that oil lamps were burned in front of each patriarch portrait in the corridor as worship. Moreover, when an offering ritual dedicated to the Forty-two *Xiansheng* 四十二賢聖 was performed in the same cloister two days after the festival, a more intricate form of this worship was given: "In front of the Buddha hall lay out forty-two portraits of *Xiansheng* and all sorts of rare colored silks beyond count. As for the countenances of the *Xiansheng*, some were concentrating with closed eyes, others with faces uplifted were gazing into the distance, others looking to the side seemed to be speaking, and others with lowered visages regarded the ground. The forty-two pictures had forty-two different types of countenances. As for the differences in their sitting postures for meditation, some sat in the full cross-legged position and others in the half cross-legged position. Their postures thus differed. Besides the Forty-two *Xiansheng*, there were paintings of Mañjuśrī and Samantabhadra and of Jīvamjīvaka and Kalavinka birds. At sunset, they lit lamps and offered them to the paintings of the saints. At night, they chanted praises, worshiped Buddha, and recited Sanskrit hymns of praise. The monks reciting Sanskrit came in together, some of them holding golden lotuses and jeweled banners, and sat in a row in front of [the pictures of] the saints and intoned together Sanskrit hymns of praise. They went through the night without resting, lighting a cup lamp in front of each saint (Reischauer 1955, pp. 72–73)".

The static nature of the postures of the priests, the offering objects, and the figure postures within the paintings characterized the Forty-two *Xiansheng* ritual. It is reasonable to assume that the participants in the Lantern festival celebration envisioned this static pattern when making offerings to the patriarch paintings in the corridor. Nevertheless, we should not overlook the incense-procession ceremony that occurred in the very same corridor (Figure 2). The Kaiyuan Monastery, extensively reconstructed in the seventh century and serving as the official Buddhist institution of the prefecture, undoubtedly had architecture designed primarily for state ceremonies.[45] While monastic architecture continued to host ceremonies of maigre feast and incense procession for centuries, religious practices developed, introducing new ideas to challenge the existing pattern of ritual-spatial use. The Forty-two *Xiansheng* offering ritual, as previously mentioned, was established by Tiantai master Zhanran during the latter half of the eighth century. As evident from Ennin's account, the performance of this ritual diverged significantly from the incense procession and transformed the manner in which patriarchs in the corridor were worshipped during non-state ceremony occasions. The use of corridors for both mobile and static ritual performances reflects a dualistic approach during the late Tang period, marking it as a transitional period in the history of Buddhist religion and architecture.

The transition that occurred in the late eighth to ninth centuries foreshadowed a significant change in post-Tang Buddhist architecture. As many studies have highlighted, a universal phenomenon in monasteries from the eleventh and twelfth centuries was the closure of the four-sided open corridors and their transformation into semi-open verandahs flanking the main image hall (G. Wang 2016, pp. 1470–71). A closer examination of monasteries built during the Khitan Liao (916–1125) and Jurchen Jin (1115–1234) dynasties further reveals that these verandah structures were actually long chambers, with narrow pillared walkways at the front, housing rows of arhat statuettes for worship (Figure 9) (Z. Xu 2016, pp. 105–16). Consequently, they are typically known as *luohan dong* 羅漢洞 (arhat grotto) in various epigraphic documents. The architectural transformation can be attributed to a variety of complex factors, which may include paradigm shifts in social, cultural, and ideological spheres during the transition between the Tang and Song dynasties, as well as the aftermath of the anti-Buddhist persecutions. However, one of the primary driving forces behind this change was likely the declining role of Buddhist religion in state rituals.

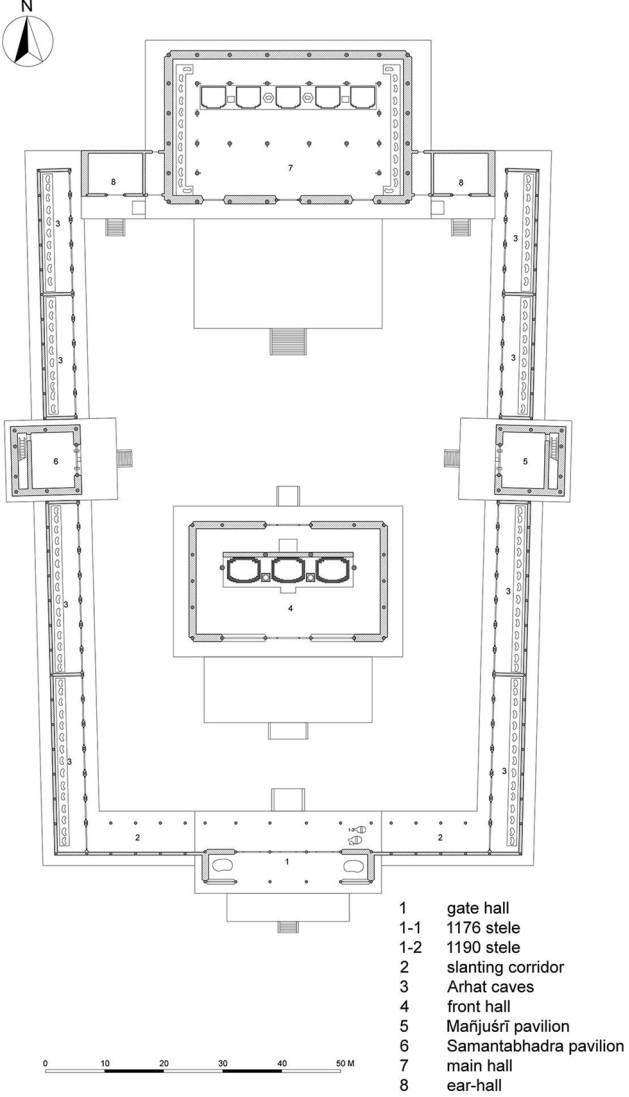

**Figure 9.** Reconstruction plan of Shanhua Monastery at 1143 (Z. Xu 2016, Figures 2–19).

| | |
|---|---|
| 1 | gate hall |
| 1-1 | 1176 stele |
| 1-2 | 1190 stele |
| 2 | slanting corridor |
| 3 | Arhat caves |
| 4 | front hall |
| 5 | Mañjuśrī pavilion |
| 6 | Samantabhadra pavilion |
| 7 | main hall |
| 8 | ear-hall |

The eleventh-century reform of *guoji xingxiang* by the Song imperial court perfectly mirrored the changing role of Buddhist monasteries in the state ritual system. In the post-Tang period, Buddhism managed to survive nationwide persecutions and continued to serve the state. *Da Song sengshi lüe* 大宋僧史略 (Abridged History of Sangha under the Song), an official history of monasticism compiled in 987, offers concise yet substantial evidence for the continuous performance of *guoji xingxiang* in Buddhist monasteries throughout the ninth and tenth centuries (Zanning 1924–1933, 241b26–242a14). However, Emperor Zhenzong 真宗 (r. 997–1022), renowned for his Taoist beliefs, commissioned the construction of Jingling Palace 景靈宮 at Qufu 曲阜 in 1012. Building this state sacrificial site was one of the emperor's efforts to restructure the Tang model and redirect the service of national mourning away from the Buddhist church. Scholars suggest that between 1032 and 1082, when holy portraits of the late emperors and empresses were transferred to Jingling Palace, the Song imperial court designated this site as the exclusive venue for *guoji xingxiang* ceremonies (J. Yang 2021; M. Wang 2016).

As Buddhist monasteries ceased to be state platforms of ceremony, their architectural spaces, which had previously been envisioned to manifest imperial authority and solemnity, gained the freedom to develop according to the needs of the monastic community and the public. In particular, the corridor and the courtyard it defined, which were used for incense processions and grand musical-dance celebrations, respectively, transitioned away from these programs that represented the imperial presence. Consequently, these spaces were no longer

perceived as places with ritual significance. Textual accounts of the Song monastic rituals mainly come from *Chanyuan qinggui* 禪苑清規 (Rules of Purity for the Chan Monastery), a Chan monastic code compiled in 1103 that gives a comprehensive set of rules for virtually every aspect of life in large public monasteries of eleventh-century China.[46] The text details the procedure for organizing a maigre feast, prescribing several designated venues, including a storage hall, a lecture hall, and a dining hall. The performance of the incense procession remained a part of the ceremony and was carried out by patrons holding a censer and circumambulating the hall where invited monks were seated (Zongze 1975–1989, 538c19–539a19). The other occasion that involves an incense procession in the *Chanyuan Qinggui* is the reading of the sūtra, which is performed within the lecture hall or library hall (Zongze 1975–1989, 538b24–c18). When the maigre feast fell outside of the intended program of the Buddhist corridor, the structure served solely as a functional passageway connecting various grand halls and as a sanctuary for worship and veneration of divine monks. As a result, closing the inner part of a corridor became essential to separate the mobile, non-ceremonial passageway from the static, cultic sanctuary. This is particularly true because the Song-dynasty individual performance of worship, marked by prostration while holding burning incense, was as static as the Forty-two *Xiansheng* offering ritual observed by Ennin in 839.[47]

## 5. Conclusions: Seeing Medieval Chinese Monastery through the Peripheral Structure

Corridors are widely recognized as a key element that characterized the architectural landscape of Tang Buddhist monasteries. This study, by foregrounding ceremonial programs as a useful tool in revealing ideas of spatial design, unfolds a series of intricate ritual-architectural interplays that transformed the Buddhist corridor from an auditorium backed by mullioned windows into a shrine for worshiping divine monks, a portrait gallery for constructing sectarian patriarchal lineage, and a platform for manifesting imperial authority. It illustrates how the inherent mobile nature of the corridor function could give rise to the new iconography of walking monk imagery. Moreover, this study strengthens the scholarly proposal linking the decline of walking monk imagery with the disuse of corridors between the 10th and 12th centuries by providing a comprehensive ritual context for the transformation (Yu 2016). It could further offer a compelling argument that the decline of iconography was not a discontinuation but rather a continuum of the old tradition as the art-architectural form transitioned from the Tang corridor paintings of walking monks to the Liao-Song verandah "caves" featuring enshrined seated *luohan* statues.

The significance of studying the Buddhist corridor lies not only in understanding the art and architectural transformation within the religion itself but also in throwing light on some previously unnoticed influence of Buddhist ideas on medieval Chinese temple architecture. In contrast to Buddhist monasteries, Chinese temple architecture has a longer history of using corridors, with the earliest textual record from the second-century Sangong Shrine 三公祠 in Yuanshi County 元氏縣 (in present-day Shijiazhuang).[48] There is no evidence to suggest that these early non-Buddhist corridors contained murals for worship. A study of Confucian temples indicates that such transformation likely occurred much later than in their Buddhist counterparts, as it highlights a significant spatial shift in Confucian temples between the Tang and Song dynasties, where the area for venerating Confucius' seventy disciples moved from the main sacrificial hall to the flanking verandahs (Shen 2015, p. 219).

In the existing discourse of Chinese architectural history, only a few studies draw attention to the issues of space and ritual. The persistent mismatch between surviving buildings and textual accounts could be an excuse for the limited focus; however, there are still plenty of fields, such as the Buddhist corridor, where materials pertinent to spatial, ceremonial, and pictorial programs are available, allowing for an in-depth examination. More than half a century ago, Japanese architectural historian Mitsuo Inoue carried out a study of Asuka and Nara period monasteries, with a keen emphasis on the dialogue between human activity and spatial construction. From this analysis, he addressed a critical transformation of the corridor (Japanese: *kairō*) and its shift in spatial connotation from a mere fence to an auditorium (Inoue 1969, pp. 61–73, 90–106). Professor Inoue's groundbreaking research serves as a foundational

influence for this article, and we aspire to establish a trajectory for subsequent scholarly works in this field.

**Funding:** This research received no external funding.

**Data Availability Statement:** No new data were created or analyzed in this study. Data sharing is not applicable to this article.

**Conflicts of Interest:** The author declares no conflicts of interest.

**Appendix A**

**Table A1.** Corridor Murals at the Sui-Tang Buddhist Monasteries.

| Monastery | Mural Location and Program | Dates of the Mural | Source |
|---|---|---|---|
| 1. Puti Monastery 菩提寺, Chang'an 長安 | Eastern Corridor | 582 AD [1] | (Duan 2015, p. 1840) |
| 2. Zhaojinggong Monastery 赵景公寺, Chang'an | Southern bays of the Eastern Corridor [of the Main Cloister] Walking monks 行僧 | 583–8th c. | (Zhang 2018, p. 77) |
| | The Southern Corridor [of the Main Cloister] | Early 8th century [2] | |
| | Western Corridor of the Sanjie Cloister 三階院 Illustration of Western Pure Land and the Sixteen Ways of Meditation 西方變及十六對事 | mid-seventh century [3] | (Duan 2015, p. 1791) |
| 3. Yongtai Monastery 永泰寺, Chang'an | Western Corridor Holy monks 聖僧 | 584 AD [4] | (Zhang 2018, p. 87) |
| 4. Jingyu Monastery 靜域寺, Chang'an | Eastern Corridor of Dhyana Cloister 禪院 Trees, rocks, and eminent monks 高僧 | 585 AD [5] | (Duan 2015, p. 1893) |
| 5. Linghua Monastery 靈華寺, Chang'an | Western Corridor Sixteen standing eminent monks 高僧, which may be accompanied by the Ten Great Disciples [of the Buddha] 十大弟子 [6] | 586–742 AD [7] | (Duan 2015, p. 1803) |
| 6. Cien Monastery 慈恩寺, Chang'an | The two Corridors of the First Cloister counting from the north off the eastern corridor of the Main Cloister | 648 AD [8] | (Zhang 2018, p. 73) |
| | The Western Corridor of the First Cloister counting from the north off the eastern corridor of the Main Cloister Walking monks 行僧 | 742–756 AD [9] | |
| 7. Yide Monastery 懿德寺, Chang'an | Eastern side of the Corridor to the west of the Gate Hall Landscape 山水 | Early 7th century [10] | (Zhang 2018, p. 83) |
| 8. Baocha Monastery 寶刹寺, Chang'an | Western Corridor Hell scenes 地獄變 | Early 7th century [11] | (Zhang 2018, p. 75) |
| 9. Shengguang Monastery 勝光寺, Chang'an | Southern Corridor | 650–683 AD [12] | (Zhang 2018, p. 84) |
| 10. Ximing Monastery 西明寺, Chang'an | Eastern Corridor Transmitters' Portraits of the Dharma 傳法者圖 including Lifang 利防 and Dharmakāla 曇柯迦羅 | 656 AD [13] | (Zhang 2018, p. 84) |
| 11. Zhaofu Monastery 招福寺, Chang'an | Long corridor Peculiar-styled Paintings | 667 AD [14] | (Duan 2015, p. 1910) |

**Table A1.** *Cont.*

| Monastery | Mural Location and Program | Dates of the Mural | Source |
|---|---|---|---|
| 12. Jing'ai Monastery 敬愛寺, Luoyang 洛陽 | Eastern and Western Gauze Corridors 紗廊 of the Main Cloister 大院 Walking monks 行僧 including Tang Sanzang 唐三藏, i.e., Xuanzang | 690–705 AD | (Zhang 2018, p. 92) |
| | Western Corridor of Dhayana Cloister 禪院 Scenes from Sūryagarbha and Candragarbha Sūtras 日藏月藏經變, and scenes showing the different rewards of karma 業報差別變 | 722 AD | (Zhang 2018, p. 91) |
| 13. Dayun Monastery 大雲寺, Wuwei 武威 | Encircling Corridors 迴廊 of the Southern Dhyana Cloister 南禪院 Portraits of Arhats and Divine Monks as the Dharma Transmitters 付法藏羅漢聖僧變, Scenes of Kāśyapa Mātaṅga and Dharmaratna's introduction of Dharma to the East 摩騰法(蘭)東來變、The Scene of Seven Maidens Avadāna tale 七女變. | 711 AD | (Zhang 2006) |
| 14. Qianfu Monastery 千福寺, Chang'an | Western Corridor of the Western Pagoda Cloister 西塔院 The portrait of the Celestial Master 天師, the portrait of the Venerable Master Chujin 楚金, and the scene of Maitreya's descent to this world 彌勒下生變 | 745 AD [15] | (Zhang 2018, pp. 80–81) |
| 15. Jianfu Monastery 薦福寺, Chang'an | Northern Corridor of the *Vinaya* Cloister 律院 | Early 8th century [16] | (Zhang 2018, p. 72) |
| | Corridor of the Southwestern Cloister Walking monks 行僧 | Early 8th century [17] | |
| 16. Xingtang Monastery 興唐寺, Chang'an | The Southern Corridor of the Pure Land Cloister 净土院 A scene from the Diamond Sūtra 金剛經變 and the Story of Empress Chi 郗后[18] and so forth | 732 AD [19] | (Zhang 2018, pp. 75–76) |
| 17. Anguo Monastery 安國寺, Chang'an | Five walls at the Corridor to the west of the Gate Hall of the Eastern Dhyana Cloister 東禪院 Eight Legions of Indra and Brahmā 釋梵八部 | 710 AD [20] | (Duan 2015, p. 1774) |
| 18. Zisheng Monastery 資聖寺, Chang'an | Northern Corner of the Western Corridor Portrait of Heavenly Maidens approaching pagoda 近塔天女 | Early 8th century [21] | (Duan 2015, p. 1925) |
| | Eastern and Western Corridors of the Guanyin Cloister 觀音院 Forty-two Holy Monks 四十二賢聖 including Nāgārjuna 龍樹 and Śāṇavāsa 商那和修 | 763–777 AD [22] | |
| 19. Xuanfa Monastery 玄法寺, Chang'an | Western Corridor A pair of pine trees 雙松 | 756–762 AD [23] | (Duan 2015, p. 1826) |
| | Eastern Corridor of Mañjuśrī Cloister 曼殊院 Elephants, horses, and congregation in the courtyard 廷下象馬人物 | ~772 AD | |
| 20. Great Shengci Monastery 大聖慈寺, Chengdu 成都 | Eastern and Western Corridors [of the Front Main Cloister] Portraits of Eminent Monks in Walking Postures 行道高僧, including Aśvaghoṣa 馬鳴 and Āryadeva 提婆 | 758 AD | (Huang 1963, p. 5) |

**Table A1.** *Cont.*

| Monastery | Mural Location and Program | Dates of the Mural | Source |
|---|---|---|---|
| 20. Great Shengci Monastery 大聖慈寺, Chengdu 成都 | Southern Corridor of the Front Main Cloister 前寺 Twenty-eight Patriarchs in Walking Posture 行道二十八祖 Northern Corridor of the Front Main Cloister 前寺 Over sixty arhats 行道羅漢 in walking posture | 826 AD [24] | (Huang 1963, p. 8) |
| | Western Corridor of the Ultimate Bliss Cloister 極樂院 Scenes from the proof of Diamond Sūtra's efficacy 金剛經驗 and Scenes from the Golden Light Sutra 金光明經變 | 826 AD [25] | |
| | Southern Corridor [of an Unknown Cloister] Seventeen Protective Deities 十七護神 including Yakṣa Generals 藥叉大將, Nāga King Vāsuki 和修吉龍王, Hārītī 鬼子母, and Heavenly Maiden 天女. | 847–879 AD [26] | (Huang 1963, p. 3) |
| 21. Baoying Monastery 寶應寺, Chang'an | Northern bays of the Western Corridor Demons and Divinities 鬼神 | 769 AD [27] | (Duan 2015, p. 1816) |
| 22. Longxing Monastery 龍興寺, Yangzhou 揚州 | Southern Corridor of the Lotus Cloister 法花院 Portraits of Master Nanyue 南岳大師 and over twenty monks who received miraculous responses by hand-copying and reciting the Lotus Sūtra | Late 8th century [28] | (Ennin 2007, pp. 90–91) |
| 23. Kaiyuan Monastery 開元寺, Yangzhou | Corridors of the Central Cloister Portraits of the Patriarchs 師影 | Between 593 and 839 AD [29] | (Ennin 2007, p. 96) |

[1] The dating is determined by considering the construction date of the monastery and the active period of the painter Zheng Fashi 鄭法士 (Sui dynasty). [2] The dating is determined by considering the active period of the painter Wu Daozi (685–758) 吳道子. [3] The dating is determined by considering the construction date of the monastery and the active period of the painter Fan Changshou范長壽 (during the reign of Emperor Gaozong). [4] The dating is determined by considering the construction date of the monastery and the active period of the painter Li Ya李雅 (Sui dynasty). [5] The date is determined by considering the construction date of the monastery. [6] The monastery is also referred to as Yunhua Monastery 雲花寺, (Ono 1989, p. 155). In Enchin's catalogue, there is a work titled "Eulogy of the Ten Great Disciples from the Yunhua Monastery in Upper Capital 上都雲花寺十大弟子贊". This document is related to the painting of the Buddha's disciples at Yunhua Monastery and is likely associated with the same corridor space mentioned in Duan Chengshi's text. See (Enchin 1924–1933, 1094c23). [7] The dating is determined by considering the construction date of the monastery and the re-installation date of the mural. [8] The dating is determined by considering the construction date of the monastery and the active period of the painter Yan Liben 閻立本 (601–673). [9] The dating is determined by considering the active period of the painter Li Guonu李果奴. [10] The dating is determined by considering the active period of the painter Chen Jingyan 陳靜眼. [11] The dating is determined by considering the active period of the painter Chen Jingyan. [12] The dating is determined by considering the active period of the painter Yin Lin 尹琳. [13] The dating is determined by considering the construction date of the monastery. [14] The dating is determined by considering the construction date of the monastery. [15] The dating is determined by considering the construction date of the Western Pagoda Precinct, which is given in Yan Zhenqing's Duobao Pagoda Stele in 752. [16] The dating is determined by considering the active period of the painter Zhang Zao張璪 and Bi Hong畢宏. [17] The dating is determined by considering the active period of the painter Wu Daozi. [18] This is very likely to be a narrative scene depicting Emperor Wudi's salvation of his deceased wife, Lady Chi, who was reborn as a python after her death. See (Fang 2021). [19] The dating is determined by considering the construction date of the monastery and the active period of the painter Wu Daozi. [20] The dating is determined by considering the construction date of the monastery and the active period of the painter monk Sidao 思道. [21] The dating is determined by considering the active period of the painter Yang Tan 楊坦. [22] The dating of the painting is determined by considering the signature of Yuan Zai (713–777), the author of the painting eulogy, who signed as Zhongshu 中書. This indicates that the painting was created between 763 and 777 during his tenure in the government position. [23] The dating is determined by considering the active period of the painter Liu Zheng 劉整. [24] The dating is determined by considering the active period of the painter Zuo Quan左全. [25] The dating is determined by considering the active period of the painter Zuo Quan. [26] The dating is determined by considering the active period of the painter Fan Qiong 范瓊. [27] The dating is determined by considering the construction date of the monastery and the active period of the painter Yang Xiuzhi 楊岫之. [28] The dating is determined by considering the active period of the painter Han Gan (706–783) 韓幹. [29] The dating is determined by considering the construction date of the monastery and the time of Ennin's record.

## Notes

[1] For related discussions, see (Greene 2013).

2   Prominent examples of palace halls with portrait paintings include the Qilin Pavilion麒麟閣of Weiyang Palace 未央宮 (dated to 51 BCE, as mentioned in *Hanshu* 漢書, fascicle 54), the Lingguang Hall 靈光殿 (with portraits dating back to the early Eastern Han, as mentioned in Wang Yanshou's *Rhapsody on Lingguang Hall of Lu Kingdom*), and the Jingfu Hall 景福殿 of Xuchang Palace 許昌宮 (dated to 232–3 AD, as mentioned in He Yan's *Rhapsody on Jinfu Hall* 景福殿賦). For governmental offices, the Eastern Han ritual text *Hanguan dianzhi yishi xuanyong* 漢官典職儀式選用 (Han officials' administrative ceremonials selected for use) documents that portraits of historical heroes were painted on the walls of the Department of State Affairs 尚書省 at Chang'an, the capital city of Western Han (see *Chuxue ji* 初學記, fascicles 11 and 24). Additionally, the Eastern Han work *Hanguan Yi* 漢官儀 (Ceremonials for Han Offices) documents the tradition of displaying portraits of senior officials on the walls of the audience halls of regional government offices. For tombs and shrines, best known and preserved is the Wuliang Shrine (built in 151 AD) in Shandong. For more extensive examination of Han mural paintings, see (Lian 2022, pp. 21–79).

3   This information is derived from a quotation purportedly originating from the fourth-century text *Yezhong ji* 鄴中記 (A Record of Ye), as cited in (Cui 1522, p. 605). S Given that the described edifice dates back to a sixth-century palace, and the Yezhong ji has only been preserved in fragmentary form, including several passages from the mid-Tang work *Yedu gushi* (Tales from the Capital of Ye 鄴都故事), it is posited that the latter text, *Yedu gushi*, serves as the veritable source for this information.

4   In medieval Chinese literature, the character *xuan* 軒 embodies a multitude of meanings, encompassing a style of chariot, a type of architecture, or an architectural element. Li Shan 李善 (630–689), an early Tang scholar, provided an elucidation of the term *xuanlang* in his commentary on *Wenxuan* 文選, characterizing it as an elongated corridor furnished with windows, or alternatively, a corridor featuring windows.

5   For the comprehensive study of the divine monk cult in medieval China, see (Liu 2013).

6   A seventh century stipulation is given in *Fayuan zhulin*, see (Daoshi 1924–1933, 610b27–c3).

7   Emperor Liang Wudi is known for ordering the compilation of Manual for Offering Food to Divine Monks (*Fan shengseng fa* 飯聖僧法) and composing eulogies on divine monk portraits, see (Liu 2013).

8   This tale is reported by Daoxuan in three separate works, including (Daoxuan 1924–1933c, 424a1–b14; 1924–1933e, 879b28-c4; 1924–1933f, 647c22–649a15). The story details are slightly different.

9   For the study of *guoji xingxiang*, see (P. Wang 2020; Nie 2015).

10   In ninth-century Chinese and Japanese literature, the term *xiang* (Japanese. *hisashi*) 廂 has two distinct interpretations. It may refer to the narrow, aisle-like interior space that surrounds the core of a building or to the long corridor that encircles a courtyard. If the "eastern, northern, and western *xiang*" were to indicate the three sides of aisles within a hall, this area should be large enough to house a congregation of five hundred monks. Based on general observations and common sense, the minimal size for an adult individual sitting on the floor is around 0.5 square meters. Therefore, it is estimated that the space needed to accommodate 500 monks would be no less than 250 square meters. The eastern hall of Foguang monastery 佛光寺 at Mt. Wutai 五臺山, which is always considered a medium-scale Tang Buddhist hall, has a usable area of 229 square meters in the three side aisles, which would be extremely crowded if five hundred monks were to sit there (The measurement of this building is found in (Zhang and Li 2010)). Moreover, studies indicate that a popular practice of *guoji xingxiang* in Tang capital monasteries involved hosting the thousand-monk-feast (*qianseng zhai* 千僧齋) (P. Wang 2020). Housing this congregation would require at least 460 square meters. Even the largest existing Buddhist hall, the Liao-dynasty main hall of Fengguo monastery 奉國寺 at Yixian 義縣, is unable to meet this requirement (The measurement of this building is found in (Jianzhu Wenhua Kaochazu 2008)). Finally, as Ennin explicitly states that the assembled monks took their food in the corridor, if they were initially seated within a building, it would indeed be quite challenging to explain when and why they left the building and relocated to the corridor. Such a noticeable movement would likely not have been overlooked by Ennin or excluded from his detailed report. In conclusion, the most plausible interpretation of the term *xiang* in this context is the corridor of the monastery.

11   In Ennin's diary, it is not explicitly stated whether the hall mentioned was the Buddha hall or the lecture hall. One may lean towards identifying it as the lecture hall because the Minister of State and Commander-in-Chief met in front of it earlier in the account. However, a recent study presents a convincing argument that the lecture hall of a Tang monastery typically did not house a Buddha image. As a result, it is more plausible to consider this structure as the Buddha hall. See (Hara 2020).

12   This is based on Alexander Soper's English translation, with several slight modifications by the author. See (Soper 1978, pp. 305–6).

13   Both the Medicine Buddha Sūtra and the Vimalakīrti Sūtra mention the practice of offering food to a group of monks and nuns, and the maigre feast scenes in the sūtra illustrations typically depict monks seated in a row within the corridor of a monastic cloister. For the maigre feast scene at Dunhuang, see (Tan 1999, pp. 191–92).

14   The study of *langxiashi* is given by (Bai 1996), and for the court audience and palace architecture of the Sui and Tang periods, see (Chen and Yi 2008).

15   There are three additional recorded events that featured the Nine-Part Music performance: the celebration of the emperor's gift of a memorial stele at the Great Ci'en Monastery in 656; the inauguration ceremony of Ximing Monastery in 658; the celebration of the emperor's gift of Buddhist images at the Zhaofu Monastery in 702. The first two events are recorded in (Huili and Yancong 1924–1933, 269a6–20 and 275c8–9), while the last event is recorded in (Duan 2015, p. 1904).

[16] The earliest performance on record is found in *Gaosengzhuan*, which includes a biography of Shaoshuo 邵碩, a divine monk active during the Liu Song (420–479) period in Sichuan. He is documented to have performed a crouching lion during the image-procession celebration of Buddha's birthday in Chengdu, (Huijiao 1924–1933, 392c25–393a7).

[17] Accounts of Jingming Monastery 景明寺, Zongsheng Monastery 宗聖寺, Changqiu Monastery 長秋寺, and Jingxing Nunnery 景興尼 寺 in *Luoyang qielan ji* reveals various entertainment forms performed during the image procession ceremony for Buddha's birthday celebrations. See (Yang 2000, pp. 35–36, 59, 64, 99).

[18] For a general introduction of Chinese court music history, see (Wang and Sun 2004).

[19] For the details of the Nine-Part Music repertory, see (Zuo 2010, pp. 93–98).

[20] Given the limitation of paintable area, however, the painter was unable to faithfully depict the entire program of the Nine-part Music performance and could only represent one band.

[21] For the study of the ceremonial plan of Tang-dynasty New Year audience, see (Guo and Shen 2022).

[22] For the study of the role of the Nine-Part Music in the New Year banquet, see (Zhou 2023).

[23] As Zhou Jing indicates, the use of the Nine-Part Music in the New Year banquet had been an established tradition by 651. See (Zhou 2023).

[24] For the study of pictorial programs of the Dayun monastery, see (Zhang 2006).

[25] Chu Suiliang and Ouyang Tong were both famed calligraphers and court officials active during the early Tang period, the short biographies of whom are found in the mid-Tang calligraphy critique *Shuduan* 書斷 (Judgments on Calligraphies).

[26] The presence of Chu Suiliang's work in Ximing Monastery is peculiar, as the politician faced demotion due to his opposition to Emperor Gaozong's proposal to make Wu Zetian 武則天 the Empress (Liu 2009). Given that the purpose of establishing Ximing Monastery in 656 was to celebrate the installation of Wu Zetian's son as the heir apparent, it remains a mystery why Emperor Gaozong and Wu Zetian (Empress Wu) would preserve Chu's calligraphic work in this monastery. This intriguing aspect lacks any scholarly insight and warrants further in-depth historical research.

[27] Existing scholarship suggests that the earliest instance of the sixteen arhats iconography, dating roughly between 586 and 742, is the portrayal of sixteen standing eminent monks on the west corridor of Linghua Monastery in Chang'an. See (Li 2010; H. Wang 1993).

[28] In Annen's catalog, compiled in 885 to include Buddhist texts and objects brought back by the eight great Japanese pilgrims, there is a painting listed from Enchin's 円珍 (814–891) collection titled "Portraits of Master Nanyue and Master Tiantai Giving a Lecture to Twenty Disciples, Collected from the Walls of Zisheng Monastery at Chang'an 長安資聖寺壁上南岳大師與天台大師等二十弟子說法 影 (Annen 1924–1933, 1132b14–15)." However, this artwork is not mentioned in the catalog that Enchin submitted to the court. Enchin's diary, Gyōrekishō 行歷抄, also suggests that the portraits of Master Nanyue and Tiantai he collected in Chang'an were actually from Qianfu Monastery 千福寺. One possible explanation for the confusion in Annen's record could be a mistake resulting from the conflation of Ennin and Enchin's records.

[29] This is known from the *Dengyō daishi shōrai daishūroku* (Saichō's Taizhou catalogue 傳教大師將來台州錄), compiled by the Japanese pilgrim Saichō (767–822) 最澄 in 804 to document Buddhist texts he collected from the Tiantai headquarter in Guoqing Monastery 國清寺 (Saichō 1924–1933, 1056a13).

[30] In Guanding's introduction to the *Mohe zhiguan*, the Dharma-Treasury Transmission lineage of twenty-three masters from Kāśyapa to Siṃha could also be reinterpreted by including Madhyāntika 末田地 as the third patriarch, resulting in a new total of twenty-four masters (Zhiyi 1924–1933, 1a13-b8).

[31] In certain manuscripts, the term *yuanta* is also transcribed as *tuanta* 團塔 (Duan 2015, p. 1926).

[32] According to the *Youyang zazu*, the corridor paintings were produced by Han Gan (706–783), with accompanying eulogy texts by Yuan Zai 元載 (713–777). Yuan Zai's signature, identified as *Zhongshu* 中書, suggests that the paintings were created between 763 and 777, during the time he held the government position of *Zhongshu shilang* 中書侍郎 under the reign of Emperor Daizong. In addition, the same source reveals that the circular pagoda features paintings of bodhisattvas by Li Zhen 李真 and paintings of flowers and birds by Bian Luan 邊鸞 (Duan 2015, p. 1925). Both artists were active during Zhenyuan period (785–805). The *Lidai minghua ji* additionally refers to Yin Lin's 尹琳 involvement in the creation of bodhisattva paintings (Zhang 2018, p. 75). Yin Lin, an artist active during Emperor Gaozong's reign, preceded Li Zhen by more than a century, making it implausible for the two to have collaborated. Nevertheless, Li Zhen was regarded as a disciple of Yin Lin and was known to mimic Yin's artistic style, which may explain their joint mention in the text (Duan 2015, p. 1908).

[33] Details of the Western Cloister Pagoda are found in (Zhang 2018, pp. 81–82) and two epigraphical sources, i.e., Ceng Xun's 岑勛 *Xijing Qianfusi Duobao fota Ganying bei* 西京千福寺多寶佛塔感應碑 (Stele of Commemorating Duobao Pogoda of Qianfu Monastery in the Western Capital [i.e., Chang'an]) and Feixi's 飛錫 *Tang guoshi Qianfusi Duobaota yuan gu fahua chujin chanshi bei* 唐國師千福寺多寶塔院故 法華楚金禪師碑 (Stele for the deceased dhyana master Fahua Chujin, the state preceptor of Tang, from Duobao Pagoda Cloister at Qianfu Monastery) (*Quan Tang wen*, juan 916).

[34] Feixi's inscription of *Tang guoshi Qianfusi Duobaota yuan gu fahua chujin chanshi bei* mentions several Prabhutaratna Pagodas were constructed by Chujin's close disciples, leading to the building of Duobao Pagodas at Wanshan Nunnery 万善尼寺 and Zijing Nunnery 資敬尼寺.

<sup>35</sup> The seven Tiantai patriarchs in the Western Pagoda Cloister must differ from the genealogical list given by Zhanran, because Xuanlang 玄朗 (673–754), the seventh patriarch in Zhanran's list, was still alive when the Cloister was completed. However, the records of Chujin's teacher and his understanding of the Tiantai lineage are unavailable, making it impossible to ascertain the details of the visual program.

<sup>36</sup> Although the display of Chujin's portrait, the founding abbot of the Western Pagoda Cloister, is understandable, the presence of *Tianshi*, or Zhang Daoling 張道陵, an Eastern Han leader of Daoism, is rather confusing. This anomaly could potentially be linked to Xuanzong's personal belief in Daoism. Scholarly research has highlighted that imperial veneration of Zhang Daoling received greater enthusiasm during Xuanzong's Tianbao era (Meyer 2006, p. 25).,

<sup>37</sup> For the location of the over twenty monk portraits, different versions of manuscripts diverge, giving two possibilities: "*menglang* (gate-corridor 門廊)" and "the same corridor (*tonglang* 同廊)." However, it is likely that both terms suggest the same location, referring to corridors on the southern side of the cloister that are connected to the gate.

<sup>38</sup> The collective title of the ten sketches is given in Annen's catalogue, composed in 885 (Annen 1924–1933, 1132b16–27). It is described as "scenes of dhyāna masters receiving miraculous responses by chanting the Lotus Sūtra 誦法花諸禪師靈異影", which perfectly corresponds with the text in Ennin's diary.

<sup>39</sup> Several monastic codes for what is forbidden during the practice of *jingxing* in corridors are given in Daoxuan's *Xinjie xinxue biqiu xinghu lüyi* 教誡新學比丘行護律儀, (Daoxuan 1924–1933d). A similar tradition is also seen in medieval Indian monasticism, (Wut 2020).

<sup>40</sup> For example, the early eighth-century story *Lanting shimoji* 蘭亭始末記 recounts that when an official visited an eastern Zhejiang monastery during the Zhenguan era (627–650), "he walked along the corridor to contemplate its murals."

<sup>41</sup> The *xingseng* image at Dayun monastery was painted by Zhou Fang 周昉, an artist active during the second half of the eighth century, see (Zhu 1985, p. 6).

<sup>42</sup> For the study of the pictorial program at Jing'ai Monastery, see (H. Wang 2006).

<sup>43</sup> The illustrations of the Diamond Sūtra discovered in Dunhuang, with the earliest example dating back to the High Tang period (704–786), depict a frontal iconic representation of the Buddha's dharma assembly. See (He 2016, pp. 99–100).

<sup>44</sup> This is based on Edwin Reischauer's English translation with several slight modifications by the author. See (Reischauer 1955, p. 71).

<sup>45</sup> The reconstruction occurred after the sack of the monastery in 623. For the history of Kaiyuan monastery, see (Daoxuan 1924–1933f, 695a6-b25).

<sup>46</sup> For the translation and study of *Chanyuan qinggui*, see (Yifa 2009).

<sup>47</sup> This practice of worship is called *shaoxiang* (burning incense 燒香) in (Zongze 1975–1989, 527b22–c2 and 534a5–7).

<sup>48</sup> The architecture of Sangong shrine is described in Sun Gai's *Sangongshan xia shenci fu* 三公山下神祠賦 (Rhapsody for the Shrine under the Sangong Mountain), see (Yan 1958, pp. 1276–77). Another textual account of pre-Sui corridor-enclosed temple compound is Xiao Gang's *Zhaozhenguan bei* 招真館碑 (Stele of Zhaozhen Taoist Monastery), which depicts a sixth-century Taoist monastery at Changshu(Yan 1958, pp. 3029–30).

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
