# Peer review of "Consecrating the Peripheral: On the Ritual, Iconographic, and Spatial Construction of Sui-Tang Buddhist Corridors"

_religions, doi:10.3390/rel15040399_

Round 1
Reviewer 1 Report
Comments and Suggestions for Authors
Please see the document

Reviewer 2 Report
Comments and Suggestions for Authors
This article makes cogent arguments regarding the historical transformation of the Buddhist corridor in Medieval Chinese Buddhist monasteries. I think all three specific arguments, the first about the state maigre feast’s influence on the functionality of the monastery corridor, the second relating to the pictorial display of dharma transmission, and the third on the correlation of the corridor and the image hall. The author provides detailed and comprehensive primary sources in the discussion, so the conclusion is well-articulated and solid.
One suggestion I would like to make is about section 2.3, which could be benefited from abridging into a more concise section. The discussion about “Nine-part music” is fascinating, yet I failed to perceive its correlation with the argument about how the imperial sponsorship influence the Buddhist ceremonial procession. The details on the musical arrangement and variations are appreciated, but a little redundant for the central argument. And sometimes the changing perspectives between the corridor and the courtyard is confusing. I would suggest trimming this section slightly, emphasizing more on the relationship between Buddhism and the state.
For line (379-380): “Entertaining performance in service of Buddhist ceremony had a long-standing tradition in India and Central Asia, and was known by the Chinese through the introduction of Buddhism”: This claim needs references to support.
Comments on the Quality of English Language
For line 769-770: "This connection is established through direct master-disciple transmission but through Nāgārjuna’s Da zhidu lun (Great Perfection of Wisdom Treatise 大智度論)": rephrasing is needed.
Reviewer 3 Report
Comments and Suggestions for Authors
An interesting subject and well researched. Some "the" and "a" inserts are needed but these are minor additions. Well written paper.
Comments on the Quality of English Language
Reviewer 4 Report
Comments and Suggestions for Authors
It is a well-researched article on the changing function of corridors in the Tang Buddhist monasteries. The author draws upon architectural, textual and historical studies to analyze the conceptual shifts underlying the transformation of the corridors' function. The analysis is compelling and supplemented by the appendix listing different monasteries and their respective corridors' scenes. The author has also reconstructed the architectural plans of the monasteries he is discussing, which helps the reader to visualize the intricacies of the arguments. The article is a great contribution to the little understood topic of corridors in the Tang monasteries.
Author Response
Thank you very much for taking the time to review this manuscript. As no revision/corrections comment is given, I will move forward, considering it satisfactory.